# Twisted bilayer zigzag-graphene nanoribbon junctions with tunable edge states

Dongfei Wang [1,6], De-Liang Bao [1,6], Qi Zheng [1], Chang-Tian Wang[1], Shiyong Wang [2], Peng Fan[1], Shantanu Mishra[2], Lei Tao[1], Yao Xiao[1], Li Huang[1], Xinliang Feng [3,4], Klaus Müllen [5], Yu-Yang Zhang [1], Roman Fasel [2], Pascal Ruffieux [2] ✉, Shixuan Du [1] ✉ & Hong-Jun Gao [1] ✉

Stacking two-dimensional layered materials such as graphene and transitional metal dichalcogenides with nonzero interlayer twist angles has recently become attractive because of the emergence of novel physical properties. Stacking of one-dimensional nanomaterials offers the lateral stacking offset as an additional parameter for modulating the resulting material properties. Here, we report that the edge states of twisted bilayer zigzag graphene nanoribbons (TBZGNRs) can be tuned with both the twist angle and the stacking offset. Strong edge state variations in the stacking region are first revealed by density functional theory (DFT) calculations. We construct and characterize twisted bilayer zigzag graphene nanoribbon (TBZGNR) systems on a Au(111) surface using scanning tunneling microscopy. A detailed analysis of three prototypical orthogonal TBZGNR junctions exhibiting different stacking offsets by means of scanning tunneling spectroscopy reveals emergent near-zero-energy states. From a comparison with DFT calculations, we conclude that the emergent edge states originate from the formation of flat bands whose energy and spin degeneracy are highly tunable with the stacking offset. Our work highlights fundamental differences between 2D and 1D twistronics and spurs further investigation of twisted one-dimensional systems.

Monolayer graphene is a zero-energy gap semimetal hosting effective massless Dirac fermions[1]. Recently, bilayer graphene with a twist angle near 1° has drawn much research attention because novel electronic ground states appear, i.e. a Mott insulating phase and superconductivity[2–8]. Electrons cannot move as freely as those in monolayer graphene due to the moiré potential and become strongly correlated. As a result, flat bands are formed near the Fermi energy. In tunnelling experiments, the flat bands reveal themselves as differential conductance peaks with near-zero energy[9–12]. However, this is not the

first time that researchers have observed flat bands in graphene systems. For example, the electrons at the edge of zigzag graphene nanoribbons (ZGNRs) become strongly correlated when the width of the ribbon decreases[13–16]. As a result, energy bands with little dispersion emerge in the range of $2\pi/3 \le |\mathbf{k}| \le \pi$ in reciprocal space (the wavenumber $\mathbf{k}$ is normalized by the primitive translation vector of the ZGNR)[17–20], corresponding to the edge states of the ZGNR. Manipulation of such edge states with tailored properties, such as antiferromagnetic semiconductor to ferromagnetic half-metal transition[20], spin-splitting

[1]Institute of Physics & University of Chinese Academy of Sciences, 100190 Beijing, China. [2]Nanotech@surfaces Laboratory, Empa, Swiss Federal Laboratories for Materials Science and Technology, Überlandstrasse 129, 8600 Dübendorf, Switzerland. [3]Center for Advancing Electronics Dresden (cfaed) and Faculty of Chemistry and Food Chemistry, Technische Universität Dresden, 01062 Dresden, Germany. [4]Max Planck Institute of Microstructure Physics, Weinberg 2, 06120 Halle, Germany. [5]Max Planck Institute for Polymer Research, Ackermannweg 10, 55128 Mainz, Germany. [6]These authors contributed equally: Dongfei Wang, De-Liang Bao. ✉e-mail: pascal.ruffieux@empa.ch; sxdu@iphy.ac.cn; hjgao@iphy.ac.cn

of dopant edge states[21] and topological order[22], is a long-lasting interesting topic with potential applications in nanodevices, i.e. spintronics[23,24] and quantum bits[25]. One of the methods used to tune the edge states involves stacking of one ZGNR on top of another in a parallel way. There, the energy gaps between the flat bands can be modulated with different sublattices matching up[26–30]. Recently, specially cut-off edges of twist bilayer graphene have been revealed to host inhomogeneous edge states[31–33]. Moreover, crossed GNRs are theoretically predicted to be beam splitters and electron mirrors when integrated into nanodevices[34–36]. All of the above results suggest new possibilities for tuning the ZGNR edge states in a bilayer case. However, pioneering experimental and theoretical research demonstrating the tunability of the edge state with both the twist angle and stacking offset is still missing.

In this paper, we demonstrate that the edge states of twisted bilayer zigzag-graphene nanoribbons (TBZGNRs) are highly tunable from both theoretical and experimental perspectives. First, modelling TBZGNR junctions with two 6-ZGNRs (the width of the ZGNR is 6 carbon atom chains) and density-functional-theory (DFT) calculations reveal that the edge states can be tuned over a wide range by changing not only the twist angles but also the in-plane stacking offset. Second, TBZGNR junctions were constructed with twist angle $\theta$ well controlled

by STM tip lateral manipulation (with accuracy less than 5° and $\theta$ between 30° and 90°). Spatially resolved scanning tunnelling spectroscopy (STS) on several edges of the orthogonal TBZGNR junctions revealed two main features: (1) a reduction in the energy gap compared to that of monolayer ZGNR, and 2) emergent near-zero-energy peaks at the edges. Additional detailed DFT calculations were performed on several TBZGNR models with $\theta = 90°$. The results showed that the emergent peaks are attributable to the formation of near-zero-energy flat bands located at the edge of the stacking area due to the interlayer interaction. Moreover, the spin degeneracy of these flat bands is highly tunable with the in-plane stacking offset, which dominates the stacking symmetry. Additional calculations suggested that the out-of-plane stacking offset (interlayer distance), whose change affects the interlayer electrostatic potential and edge spin distribution, is another parameter with which to manipulate the overlapping edge states.

## Results

### Tunability of edge states revealed by DFT calculations

The edge states of monolayer ZGNRs manifest themselves as dispersionless bands and present as van-Hove singularities (VHS) in the calculated density of states (DOS) (peaks of the grey shadow in Fig. 1b–e). A band gap close to the Fermi energy develops due to

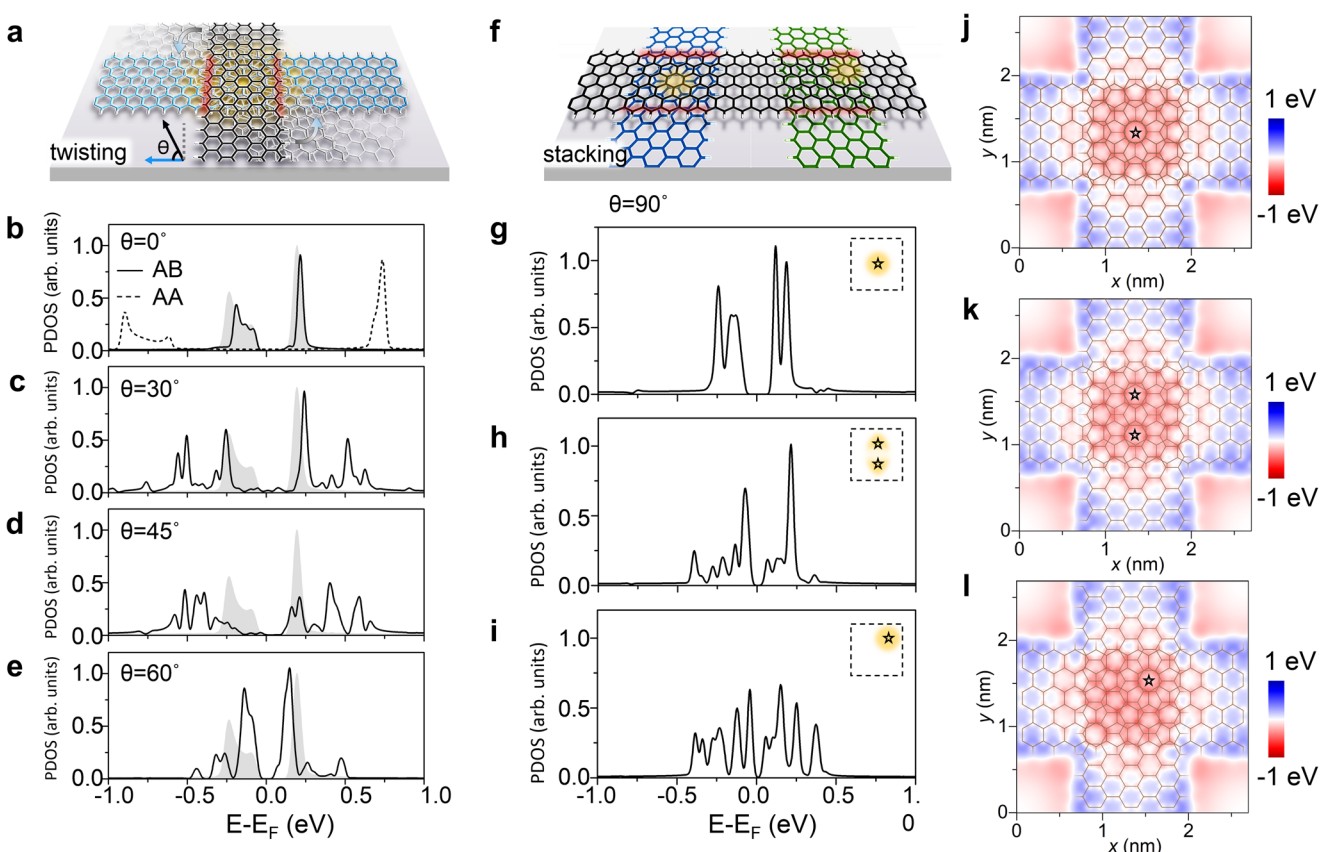

**Fig. 1 | Tunability of zigzag graphene nanoribbon (ZGNR) edge states with twist angles and in-plane stacking offsets. a** Schematic of twisted bilayer zigzag graphene nanoribbon (TBZGNR) junctions with varying twist angles. Blue and black ribbons represent bottom and top layer ZGNRs, respectively (also in **f**). The angle $\theta$ represents the twist angle between the top and the bottom ribbon. Red shadow regions illustrate the edges of the top layer ZGNR within the overlapping region (also in **f**). **b–e** Density-functional-theory (DFT)-calculated projected density of states (PDOS) on the edge atoms in the red shadow regions in cases with several typical twist angles. In the case of a twist angle of 0°, both $\beta$-AB (solid curve) and AA stacking (dashed curve) are considered. The grey shading represents the PDOS for edge atoms in monolayer ZGNR. The results in **c–e** are based on structures with

overlapping central hexagons, which are the most symmetric junctions. **f** Schematic of TBZGNR junctions with the same twist angle of 90° but different in-plane stacking offsets. Two typical stacking geometries are shown here for example. The yellow shadows highlight the moiré sites with AA stacking used for distinguishing different stacking configurations. **g–i** DFT-calculated PDOS of the edge atoms (within the red shadowed regions shown in **f**) in three typical TBZGNR stacking symmetries with the same twist angle of 90°. Insets show where the moiré sites are located. **j–l** DFT-calculated interlayer electrostatic potential (in the middle plane between the top and bottom GNRs) of three 90°-TBZGNRs with the atomic stackings shown in **g–i**. Here the interlayer distance is fixed at 3.0 Å for the calculation.

enhanced electron-electron interactions in finite one-dimensional (1D) geometry[13]. By placing one ribbon on top of the other, as illustrated in Fig. 1a, f, the edge states are affected largely by different twist angles and in-plane stacking offsets. When the twist angle $\theta = 0°$ (parallel), two layers of ZGNRs typically exhibit AA or AB stacking. For AA stacking, hybridization between edge states of the top and bottom ribbons is maximized, leading to a strong edge electron hopping between the ribbons. A DFT-calculated projected density of states (PDOS) on the edges of AA-stacking bilayer ZGNRs showed that the rearranged flat bands were also revealed as VHS but with a relatively larger energy gap than the monolayer case (dashed curve in Fig. 1b). In contrast, when the two ribbons achieved AB stacking, the edge states of the top and bottom ZGNRs fell in a hybridization-avoiding geometry. The flat bands barely changed[26], while the gap between them was reduced slightly (solid curve in Fig. 1b). In addition to parallel AA and AB stacking, one can also stack ZGNRs with an arbitrary twist angle $\theta$ and form a TBZGNR junction, as illustrated in Fig. 1a. Figure 1c–e clearly show that the edge states (solid curve, shown as VHS in PDOS) of TBZGNR junctions shift towards zero energy with an increasing twist angle $\theta$ when the moiré site locates at the junction centre.

It is noteworthy that for a given single twist angle, there remains a rich diversity of twist symmetries tuned by the in-plane stacking offset. This is in marked contrast to two-dimensional (2D) materials in which the twist angle alone entirely defines the moiré unit cell and hence the full stacking geometry. In relation to this, the additional in-plane stacking offset used to define the geometry of the TBZGNR can be regarded as an offset vector defining the portion of the 2D moiré unit cell describing the finite overlap area and edge segments of twisted 1D structures. As an illustrative example, a TBZGNR junction with $\theta = 90°$ (Fig. 1f) can adopt either high (left) or low (right) stacking symmetry, whereby the edge states of junctions with different symmetries show significant changes (Fig. 1g–i). Furthermore, a reduction in the stacking symmetry directly reduces the symmetry of the interlayer electrostatic potential, as shown in Fig. 1j–l. Since a lateral external electric field was predicted to alter the spin-polarized edge states of ZGNRs[20], the stacking offset-dependent electrostatic potential could be the factor altering the edge states in those $\theta = 90°$ junctions. Calculated PDOS on edge atoms in narrower 4-ZGNRs and wider 8-ZGNRs are shown in Supplementary Fig. 13 suggesting that the wider ZGNRs will produce more complicated overlapped configurations and more abundant edge states. Following the theoretical predictions described above, we built experimental TBZGNR junctions and took corresponding measurements as discussed in the following sections.

## Fabrication of TBZGNRs junction with peculiar edge state

High-quality monolayer 6-ZGNRs were synthesized on Au (111) via a bottom-up method[37] (Fig. 2a, also see the "Methods" section and Supplementary Fig. 1). It is challenging to build a TBZGNR junction directly with vertical STM tip manipulation. However, we noticed that the ribbon could easily be moved or even bent[38] on the Au surface with the STM tip (Supplementary Fig. 2). Thus, we built the junction by pushing one ribbon on top of another, which was near the step edge on the lower terrace. This idea is illustrated in Fig. 2b, c, in which a TBZGNR junction is formed with a twist angle $\theta$. The twist angle can be controlled during manipulation. We succeeded in building TBZGNR junctions with different twist angles $\theta$, as shown in Fig. 2d–f and Supplementary Fig. 3. The decoupling effect of the bottom ZGNR makes the edge states of the top ZGNR visible only in the overlap region. From Fig. 2h, one can see that the STS at the edge of the monolayer ZGNR still mimics the line shape of Au (111), but the DOS at the edge of the bilayer junction changes considerably and exhibits a pronounced peak near zero energy. The corresponding d$I$/d$V$ mapping image shown in Fig. 2g clearly revealed that this near-zero-energy peak was only localized at the TBZGNR junction edge. Once TBZGNR junctions were built, further manipulations on the top

ribbon can still be achieved in both directions relative to the bottom ribbon, as demonstrated in Fig. 2i–l.

To obtain more information regarding the edge states of the top layer ZGNR within the junction, we took the spatially resolved STS at the edges. Three typical spectra recorded at the edges of TBZGNR junctions named A, B and C, and with similar $\theta \approx 90°$, are shown in Fig. 3a–c, respectively (see STS data for the other twist angles in Supplementary Fig. 6). From Fig. 3a, we determined that the lower edge gave energy gap values of $\Delta^0 = 0.90$ eV and $\Delta^1 = 1.15$ eV ($\Delta^0$ and $\Delta^1$ denote the direct band gap and the energy gap at the Brillouin zone boundary[13]), while the upper edge gave similar gap values of $\Delta^0 = 1.07$ eV and $\Delta^1 = 1.34$ eV. Compared to the gap values[37] for the same type of ZGNR decoupled by a NaCl layer, $\Delta^0 = 1.5$ eV and $\Delta^1 = 1.9$ eV, the band gap in our case has diminished considerably. We attribute this band gap reduction to the energy bands renormalization mainly caused by the interlayer electron hopping-induced charge redistribution between the two ZGNR layers, which did not occur when the ribbon was decoupled by a NaCl layer. DFT calculations showed that in the overlapped region of TBZGNR, electrons tended to accumulate at the interface (Supplementary Fig. 4). As a result, the electron charge density at the ribbon edges was reduced, as was the corresponding effective Coulomb repulsion. The band gap reduction was proven by DFT calculations for structure Model A, as shown in Fig. 3d (see Fig. 4c for the atomic configuration). Compared to the PDOS on the edges of monolayer pristine ZGNR (grey shade), the band gaps of Model A were reduced (red and blue curves). It is worth noting that DFT calculations underestimated the band gaps, so the absolute values of the band gaps are not comparable to the experimental values. However, the relative values from the calculations are meaningful. In addition, we can not easily exclude further bandgap renormalization mechanism such as Thomas–Fermi screening when including the effect of the Au (111) surface[39].

Emergent near-zero-energy STS peaks were discovered at the edges of the other TNBZGNR junctions B and C (Fig. 3b, c). For junction B, the near-zero-energy peak existed along the whole edge, as indicated by the red dashed line in Fig. 3b (see also Fig. 2g). However, for junction C, this near-zero-energy peak was only found to lie near one corner of the junction (point 1) and decayed very fast to the other corner (Fig. 3c). In addition to the near-zero-energy peak, we also identified other peaks at positive energies, as indicated by the black dashed lines in Fig. 3b, c. Interestingly, our DFT calculations for the other structural Models B and C, as shown in Fig. 3e, f correspondingly (see Fig. 4d, e for detailed configurations), showed results similar to those of the experiments. The PDOS shown in Fig. 3e clearly shows that in Model B, the near-zero-energy peak (indicated by the red dashed line) extended along the edge from point 1 to point 5 with a slight intensity reduction at the corner. Moreover, this peak in Model C decayed rapidly from one corner to the other (Fig. 3f). Additionally, the other calculated peaks indicated by the black dashed lines in Fig. 3e, f also matched the experimental data qualitatively along the edge for both junctions B and C. It is noteworthy that the Au (111) step edges did not show any DOS anomaly near zero energy, as shown in Supplementary Fig. 7, and thus did not cause additional difficulty in the corresponding analysis. Other mechanism which can cause the DOS anomaly near zero energy such as defect state can also be ruled out (Supplementary Note 8).

## Manipulating the edge state by tuning stacking offsets

The primary difference among Models A, B and C is that their in-plane stacking symmetries, which are tuned by the in-plane stacking offset, reduced gradually (Fig. 4c–e). Model A has inversion, mirror, C2, and C6 symmetry within the overlapped region, while C6 symmetry is absent for Model B. Finally, there is no lattice symmetry for Model C. To obtain a deeper understanding of the edge states in Models A, B and C, we calculated the full band structures of the three models first, as

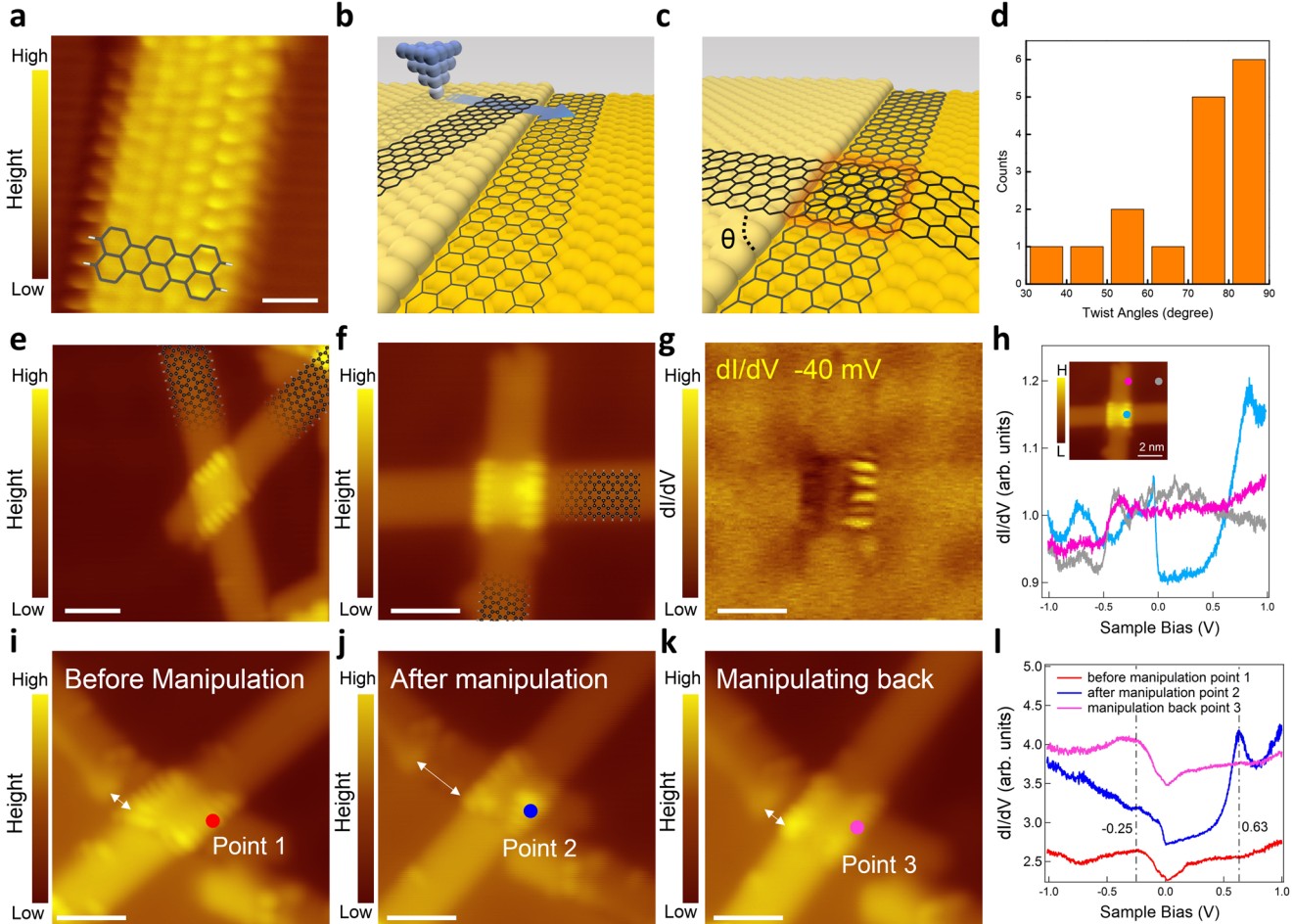

**Fig. 2 | TBZGNR junctions obtained by scanning tunnelling microscopy (STM) lateral tip manipulation.** **a** High-resolution STM topography image of the as-grown monolayer ZGNR. **b, c** Schematic diagrams of ZGNRs near a step edge before and after STM tip manipulation, respectively. **d** A histogram showing experimentally achieved twist angles $\theta$ between the top and bottom ZGNRs. **e, f** STM topography images of two as-fabricated TBZGNR junctions with the edge states of the top ribbon clearly visualized. The twist angles of these two junctions are 53° and 87°, respectively. **g** d*I*/d*V* mapping image of −40 mV of the TBZGNR junction shown in **f**. **h** Three typical scanning tunnelling spectroscopy (STS) measured on the junction edge (blue), on the edge of monolayer ZGNR (pink) and on the Au (111) surface (grey). Inset: Same STM topography image as **f** indicates where the STS were taken. **i–k** Three STM topography images demonstrating the manipulation of the top ZGNR on the surface of the bottom ZGNR. The relative motion of the top ribbon is highlighted by the white arrows. **l** Corresponding d*I*/d*V* spectra taken at points 1–3 before manipulation (red), after manipulation (blue) and manipulating the ribbon back to the initial position (pink). The vertical dashed lines highlight the change of d*I*/d*V* signals at different staking configurations. The blue and pink curves have offsets of 0.7 and 1.4 compared to the red curve for better data visualization. Scale bar: **a** 0.6 nm, **e–g**, **i–k** 2 nm. Tunnelling parameters: **a** $V$ = −0.4 V, $I$ = 620 pA; **e, i–k** $V$ = −0.3 V, $I$ = 1.0 nA; **f** $V$ = −93.7 mV, $I$ = 165 pA; **g, h** $V_{stab}$ = −0.32 V, $I_{stab}$ = 1.0 nA. $V_{osc}$ = 0.5 mV; **l** $V_{stab}$ = −0.3 V, $I_{stab}$ = 1.0 nA. $V_{osc}$ = 0.7 mV.

shown in Fig. 4f–h (solid lines). For comparison, we also calculated the band structure of a monolayer ZGNR, as shown in Fig. 4b (a justification for the calculated supercell size is presented in the Supplementary Note 7 and Supplementary Figs. 8 and 9). In the energy window we plotted, we identified 4 spin-degenerated bands below the Fermi energy for the monolayer ZGNR (Fig. 4b) as a result of band folding (see the "Methods" section). These bands were doubled to 8 but were still spin-degenerate for Models A and B because of symmetry protection (Fig. 4f, g). However, the bands below zero energy for Model C showed clear spin splitting (Fig. 4h), which was a direct result of the broken sublattice symmetry.

The band structures with projections on the upper and lower edges of the top ZGNR are superimposed on the full band structures (circles in Fig. 4b, f–h). Compared to the pristine monolayer ZGNR (Fig. 4b), we can see that the edge states (focusing on bands between −0.15 and 0 eV) in Model A were still spin degenerate but became more dispersive (Fig. 4f). With lower symmetry in Model B, the edge states kept the spin degeneracy but became isolated and closer to the Fermi energy (Fig. 4g), which explained the strong near-zero-energy peak in

the calculated PDOS. For Model C without any symmetry, spin splitting of the edge states became evident immediately. New spin-polarized flat bands were developed close to the Fermi energy (Fig. 4h). As the time reversal symmetry was still reserved in Model C, spin–orbital coupling[40] and pseudomagnetic field effects[41–43] did not lead to spin splitting at the Γ point. Other effects, such as Au step edge state, out-of-plane bending and lattice distortion, were also excluded (see Supplementary Notes 5 and 6 and Supplementary Figs. 7 and 10).

It has been indicated that an in-plane external electric field can lift the spin degeneracy of the ZGNR edge states[20]. Considering the asymmetric stacking configuration in Model C, where the moiré site is located on the corner of the junction, the effective electron charge density showed an inhomogeneous distribution within the overlap region. Thus, an inhomogeneous electrostatic potential was introduced between the edges of the top ribbon (as shown in Fig. 1l) and played the same role as an external electric field. By extracting the potential difference within the overlapped region from Fig. 1l, the estimated differential electric field is -0.05 V/Å, which is comparable with that predicted in ref. [20]. In fact, a similar asymmetric interlayer

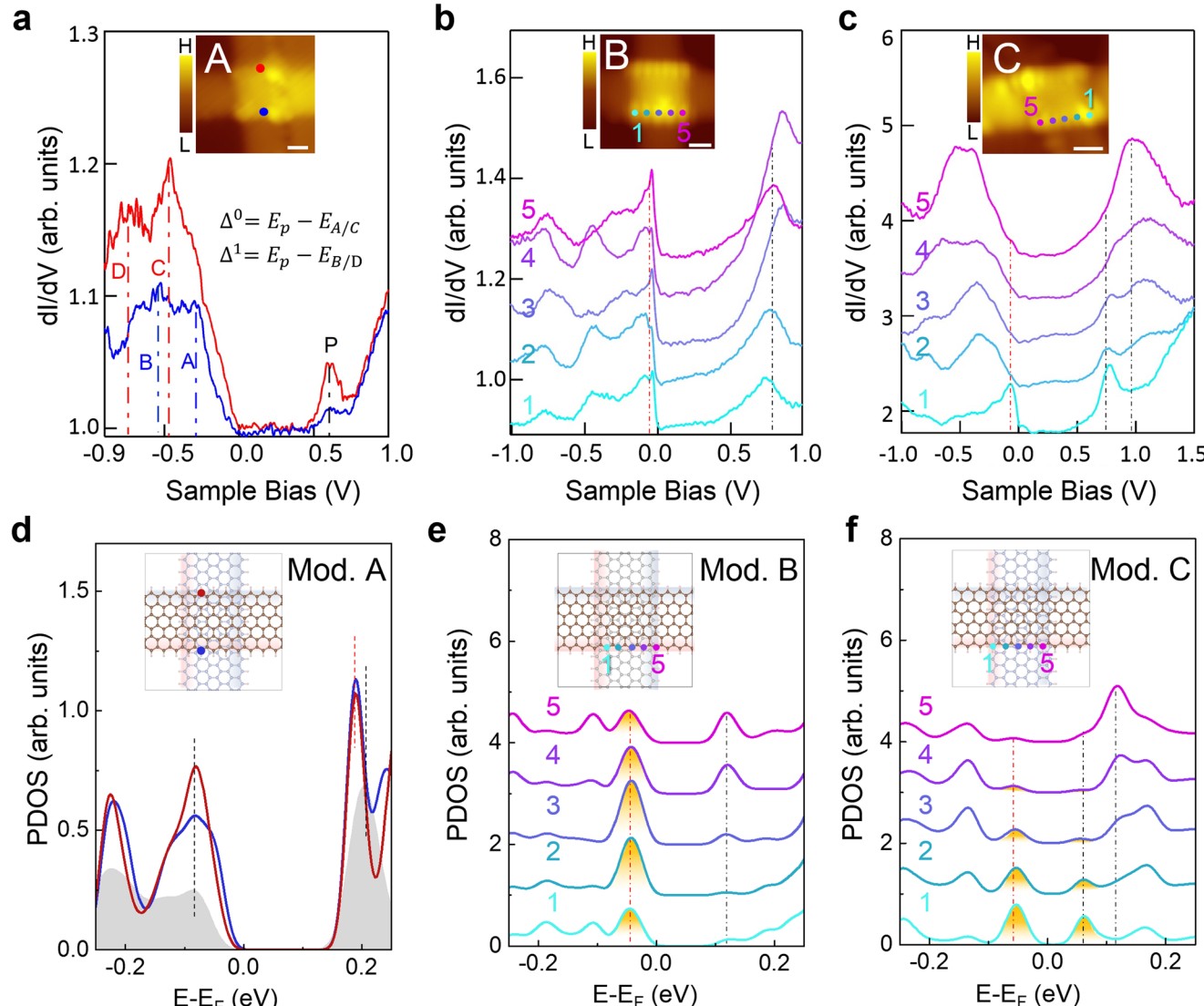

**Fig. 3 | Experimental and DFT calculated results of the edge states of 3 TBZGNR junctions with θ ≈ 90°. a–c** STS taken at the zigzag edges of three TBZGNR junctions. Insets in **a–c** are the STM images of the three junctions indicating where the STS were taken. Scale bar in insets: **a** 0.7 nm, **b** 0.76 nm, **c** 0.74 nm. The d$I$/d$V$ signals only show a gap-like feature at the edge of junction A, while a pronounced peak near zero energy was shown in TBZGNR junctions B and C (as indicated by the red vertical dashed lines). The pronounced peak near zero energy distributed along the whole bottom edge of the TBZGNR junction B, as shown in **b** while distributed only in the vicinity of the corner of junction C, as shown in (**c**). The black vertical dashed lines highlight the STS peak positions above zero energy. **d–f** DFT-calculated PDOS for the edge atoms of the three TBZGNR Models A, B and C. The red and blue curves

in **d** are the PDOS at the red and blue points shown in the inset. The grey shaded area represents the PDOS of the edge atoms in monolayer ZGNR. The curves labelled 1–5 in **e**, **f** are the PDOS for corresponding atoms 1–5 in the inset for Model B and Model C, respectively. The yellow-shaded areas highlight the PDOS peaks near zero energy. The red and black dashed lines indicate the peak energy position below and above zero energy correspondingly. The ribbon lying horizontally stands for the "top" ribbon in **d–f**. All models were structurally relaxed. The interlayer distances were optimized to be -3 Å. Tunnelling parameters: **a** $V_{stab}$ = −0.3 V, $I_{stab}$ = 1.1 nA, $V_{osc}$ = 0.5 mV; **b** $V_{stab}$ = −0.3 V, $I_{stab}$ = 1.0 nA, $V_{osc}$ = 0.5 mV; **c** $V_{stab}$ = −0.3 V, $I_{stab}$ = 1.0 nA, $V_{osc}$ = 0.7 mV.

electrostatic potential was indeed reported for crossing armchair GNRs[34]. The symmetry-reduction-induced spin splitting in edge states was double-checked by DFT calculations based on another asymmetric TBZGNR structure, Model D (Supplementary Fig. 5). It showed results consistent with those of Model C, i.e. the spin degeneracy was lifted. Based on further nc-AFM measurements on an asymmetric TBZGNR structure with twist angle 76°, the atomic structure of the junction was determined in an unambiguous way. The measured dI/dV signal across the junction also matches with the calculated PDOS using the same atomic model (Supplementary Fig. 11). At this point, we concluded the in-plane stacking offset difference is the most probable factor causing different edge states for the three orthogonal TBZGNR junctions as shown in Fig. 3a–c. Twist angles other than 90° and other widths of

GNRs produce longer or shorter overlapped edges, where we believe the symmetry on interlayer electrostatic potential still affects the edge states. However, the quantitatively calculational and experimental measurements need further explorations.

It is noteworthy that the interlayer electrostatic potential was sensitive to the distance between the two layers of TBZGNRs. As shown in Fig. 5a, when the interlayer distance was increased to 3.5 Å and beyond, the potential was weakened rapidly. The edge states of TBZGNR junctions with non-90° twist angles still need further exploration, and two of them are shown in Supplementary Fig. 6. Due to the rich array of possible stacking configurations, it will probably be very difficult to obtain systematic conclusions without knowing the atomic structure of the overlapped region.

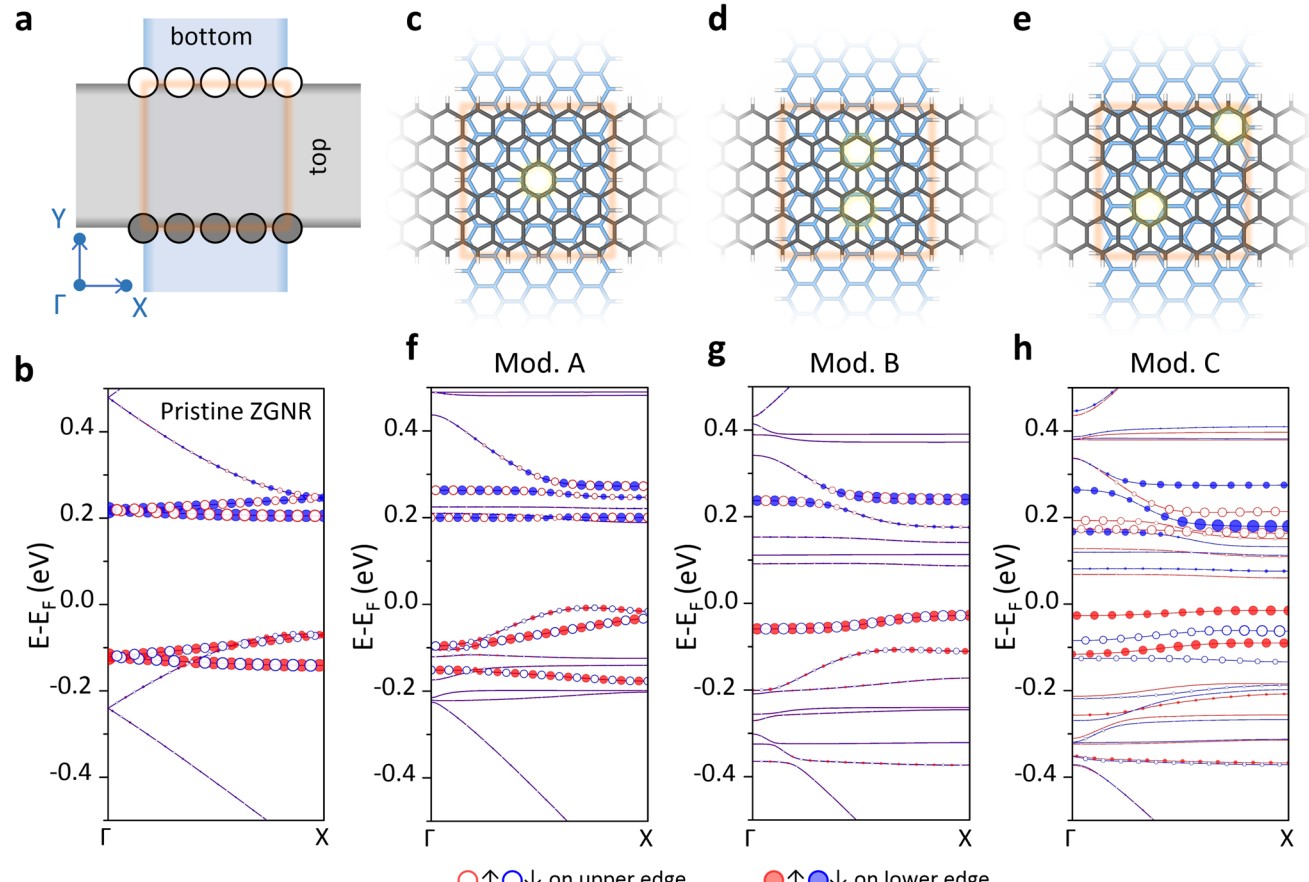

**Fig. 4 | DFT calculated band structures for three designed TBZGNR models with distinct stacking symmetries. a** Illustration of the construction of TBZGNR models. Grey and blue ribbons represent the top and bottom ZGNRs, respectively. The light red square marks the overlapped region. The open and filled circles mark the upper- and lower-edge atoms to project on, respectively. **b** Calculated band structures for the 11 × 1-supercell monolayer of pristine ZGNR. **c**–**e** Atomic structures of Models A (**c**), B (**d**), and C (**e**), showing the different stacking symmetries. The structures in grey are the top ZGNRs. The light-yellow shadow highlights the moiré sites. **f**–**h** Band structures calculated for Models A, B, and C, respectively. Edge-atom projections are represented as corresponding open/filled circles. Red and blue colours correspond to spin ↑ and spin ↓, respectively.

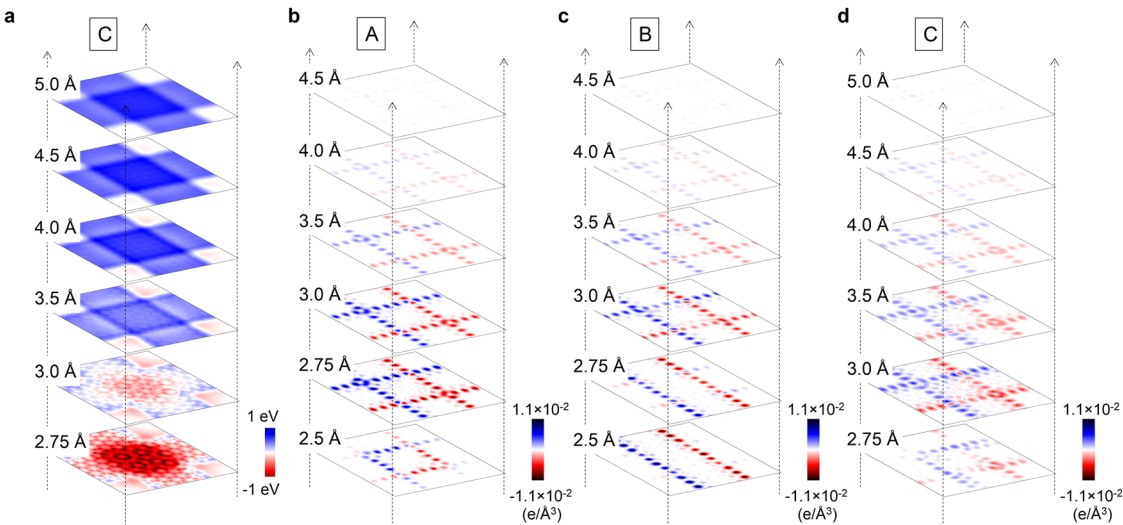

**Fig. 5 | Calculated electrostatic potential distribution and spin density distribution. a** Electrostatic potential distribution in the middle plane between TBZGNRs as a function of interlayer distance (based on Model C). Blue and red colours correspond to positive and negative values, respectively. **b**–**d** Spin density distribution in the middle plane between bilayer twist ZGNR as a function of interlayer distance. Blue and red represent majority spin and minority spin, respectively.

Considering the possible application of the TBZGNR network to spintronics, it is always worth knowing how the spin arrangements on the overlapping edges evolve when the interlayer distance changes (out-of-plane stacking offset). Starting from an initial antiferromagnetic order for each ZGNR, when the interlayer distances were larger than the optimal distance (~3.0 Å), none of the spin arrangements in any model changed, but the intensity weakened gradually (Fig. 5b–d). When the interlayer distance decreased, the spin arrangement in Model A maintained antiferromagnetic order at 2.75 Å but showed spin confinement at 2.5 Å, i.e. two antiferromagnetic corners in the overlap region (Fig. 5b). In Model B, antiferromagnetic order of only one ZGNR was demonstrated at shorter distances (Fig. 5c). In Model C, the spin arrangement became rather asymmetric at 2.75 Å and was not localized on the edges but extended into the inside of the overlapped region at 3 Å, as displayed in Fig. 5d. The above results again emphasize the significance of stacking offset and suggest that spin frustration may exist between the two layers of TBZGNRs with short interlayer distances. Therefore, the out-of-plane stacking offset can serve as a selectable parameter for tuning the edge states of the overlapped region beyond the twist angle and in-plane stacking offset when designing a spintronic device using TBZGNRs. One possible experimental realization is fabricating TBZGNR junctions encapsulated between two insulating layers, i.e. boron nitrides. The out-of-plane stacking offset can be adjusted by tuning the insulating layers.

## Discussion

We have demonstrated the presence of highly tunable edge states in the TBZGNR junction from both first-principles calculations and experiments. The featured edge states of the as-fabricated TBZGNRs combined with the reproduced theoretical results enabled us to elucidate the dominant role of stacking offsets on the edge states of TBZGNRs. Our results revealed that in twisted bilayer 1D systems, in addition to the twist angle, which is the prime factor in the 2D case, the stacking offset is another important parameter influencing the edge states as well as the charge and spin distributions of the junctions. The as-investigated 1D twisted junctions are foreseen to be construction units for nano devices, such as spin filters[23,36,44]. Our discovery also offers intriguing opportunities for explorations on 1D twisted junctions based on materials with more abundant electronic, optical, and topological properties.

## Methods

### DFT calculations

DFT calculations were performed using the Vienna ab initio simulation package (VASP)[45,46] code with the projector augmented wave (PAW)[47] method. The local spin density approximation (LSDA)[48] of Perdew–Zunger was adopted for the exchange–correlation functional. The energy cut-off of the plane-wave basis sets was 400 eV. The computational models comprised a 2.69 nm × 2.69 nm × 2 nm unit cell containing two overlapping ZGNRs with twist angles of 90°. The calculations of monolayer ZGNR used a unit cell of the same size but with only one layer of ZGNR. The numbers of carbon rings in widths/lengths of all ZGNRs were 6/11. The thickness of the vacuum layer was ~1.7 nm. The Brillouin zone was sampled with only the Γ-point. During structural relaxation, all atoms were relaxed until the force on each atom was less than 0.01 eV/Å. For the PDOS calculations, we used a $40 \times 1 \times 1$ k mesh, where 40 was along the direction of the ZGNR that was projected on.

### STM/STS manipulation and characterisation

All STM/STS measurements were performed in ultrahigh vacuum at a temperature of 4.4 K. Before switching off the feedback loop to record the differential tunnelling conductance (d$I$/d$V$) spectra, the tip was stabilized at a current ($I_{stab}$) and a sample bias voltage ($V_{bias}$). The d$I$/d$V$ signal was then recorded using a lock-in technique with a bias modulation frequency of 987 Hz. Lateral STM tip manipulation of the nanoribbon was achieved via three steps. Step 1: the target GNR was located on the upper terrace of Au(111) with another GNR close to the step edge checked with STM topography images. The manipulation path, direction and position the tip were selected near the target GNR edge, which was on the opposite of the manipulation direction. Step 2: the tip was moved closer to the Au (111) surface by adjusting the current setpoint and sample bias and typical parameters such as $V = 10$ mV and $I = 1$ nA. The feedback loop was opened and the tip was moved along the designed path with slow speed. Step 3: the nanoribbon was checked after manipulation by scanning the target area again. If the twist angle was not what the experiment required, Steps 1 and 2 were repeated until the TBZGNR junction was fabricated.

### AFM characterisation

The bond-resolved images were carried out in a Createc low-temperature STM/nc-AFM system in ultra-high vacuum (with a base pressure better than $2.0 \times 10^{-10}$ mbar). The measurements were conducted at 4.5 K. A qPlus sensor ($Q$ factor = 20,000, resonant frequency=29 kHz) with Pt-Ir tip was used for STM/nc-AFM measurements. STM characterisations were performed in constant current mode and the bias refer to the voltage on samples with respect to the tip. Nc-AFM data were taken with CO functionalized tip in constant height mode with oscillation amplitude of 100 pm.

### Sample preparation

Graphene nanoribbons were synthesised by following the growth protocol presented in a previous report[37]. After precursor deposition at room temperature with the Au(111) surface held at room temperature (Supplementary Fig. 1a), polymerization of these precursors was achieved by direct filament heating with a 2.2 A current for 10 min, and a temperature of approximately 140 °C was measured (the temperature was recorded with Optris thermometer and the emissivity was set to 0.17). These polymers were further planarized by heating again at approximately 180 °C for 10 minutes, as shown in Supplementary Fig. 1b. Most of the ribbons were synthesized on the Au terrace, with some others formed near step edges, as shown in the inset of Supplementary Fig. 1b. The width of the ribbon was approximately 1.2 nm, and the length of the ribbon was usually between 10 and 40 nm, as shown in Supplementary Fig. 1c. An atomically resolved STM topography image revealed that the monolayer ZGNR was composed of 6 zigzag carbon chains, as displayed in Fig. 2a. Due to the itinerant $d$ electron on the Au (111) surface, the intrinsic density of states of the nanoribbon was usually immersed in the Au surface state. As shown in Supplementary Fig. 1d, the d$I$/d$V$ spectra taken at the upper and lower edges of the monolayer ZGNR (red and blue curves, respectively) showed line shapes similar to those taken on the Au (111) substrate (dashed grey curve). The reduction in the density of states near −0.5 V was due to partial screening of the Au surface state by the nanoribbon. To measure the intrinsic band structure of the nanoribbon, a less conducting layer should be intercalated between the Au surface and the nanoribbon[49–51]. A previous report[37] observed the intrinsic density of states of the zigzag edge after intercalation of a NaCl layer at the ZGNR/Au interface. However, using a STM tip to achieve vertical manipulation of the ribbon and move it onto a NaCl island is rather difficult[52].

## Data availability

Relevant data supporting the key findings of this study are available within the article and in the Supplementary Information/Source data file. Additional data generated during the current study are available from the corresponding authors upon request. Source data are provided with this paper.

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

## Acknowledgements

We would like to thank Min Ouyang, Steven Louie, Sokrates T. Pantelides, and Jiebin Peng for useful discussions. H.-J.G. and S.D. gratefully acknowledge funding by National Natural Science Foundation of China (61888102). P.R. and R.F. gratefully acknowledge funding by the Swiss National Science Foundation under Grant No. IZLCZ2_170184. S.D. gratefully acknowledge funding by the Strategic Priority Research Program of Chinese Academy of Sciences (No. XDB30000000). Y.-Y.Z. gratefully acknowledge funding by the National Key Research and Development Program of China (No. 2019YFA0308500). S.D. and Y.-Y.Z. gratefully acknowledge funding by the K. C. Wong Education Foundation and the International Partnership Program of Chinese Academy of Sciences (No. 112111KYSB20160061). D.-L.B. gratefully acknowledge funding by China Postdoctoral Science Foundation (No. 2018M641511). X.F. gratefully acknowledge funding by EU Graphene Flagship 881603 (GrapheneCore3). D.-L.B., C.-T.W., L.T., Y.-Y.Z. and S.D. gratefully acknowledge the computational resources provided by the National Supercomputing Center in Tianjin.

## Author contributions

H.-J.G. supervised the overall research. S.D., P.R., R.F. and H.-J.G. designed the experiments. D.W., P.F., S.M. and S.W. performed the scanning tunneling microscopy experiments. Q.Z., Y.X., and L.H. performed the AFM experiments. D.-L.B., C.-T.W., L.T., Y.-Y.Z. and S.D. performed the theoretical calculations. X.F. and K.M. provided the molecular precursors. D.W., D.-L.B., S.D., P.R., R.F. and H.-J.G. wrote the manuscript. All authors analysed the data and contributed to the preparation of the manuscript.

## Competing interests

The authors declare no competing interests.
