## [Peer Review File · Nature Communications]

Twisted bilayer zigzag-graphene nanoribbon junctions with tunable edge statesEditorial Note: Figure S4 on page 31 in this Peer Review File is reproduced with permission from Springer Nature.

Ruffieux, P., Wang, S., Yang, B. et al. On-surface synthesis of graphene nanoribbons with zigzag edge topology. *Nature* 531, 489–492 (2016).
<https://doi.org/10.1038/nature17151>

REVIEWER COMMENTS

Reviewer #1 (Remarks to the Author):

In this work, the authors employ DFT and STM to investigate junctions of stacking zigzag graphene nanoribbons (ZGNRs) with various twist angles. The junctions were constructed by manipulating ZGNRs by STM. The main results include observation of edge states at the edges of the junctions, which are compared with DFT calculations. The experimental realization of the junctions is novel, and the results are interesting. The comparison between the measured STS and the DFT calculations seems reasonable. However, the title of the manuscript is a little confusing and misleading since the work is on the junctions or the overlapped parts of two nanoribbons. The title sounds like the studied systems are the edges of bilayer nanoribbons made from twisted bilayer graphene. In principle, the junctions are closer to a quasi-zero dimensional system than a one-dimensional one. So, the title should be changed to be more accurate. Some other detailed comments and questions are also listed below. If the authors can address the comments and questions, I suggest publishing this manuscript in Nature Communications after revision.

- 1. From the STM images in the manuscript, the stacking order (AA or AB) of the junctions can not be determined. Atomically resolved STM images of the junctions are needed to unambiguously determine the stacking order.**
- 2. For 0 twist angle (or parallel) bilayer GNRs, what are the possible edge configurations? How does the edge-to-edge interaction change the geometry of the edges?**
- 3. In Fig. 2i and Fig. 2k, the twist angles appear different. Did the manipulation change the twist angle? If so, the manipulation process is not reversible.**
- 4. The different parts of the overlapped edges of the top GNRs show different edge states, e.g., middle of the edge vs. corners of the junction, as shown in Figs. 3b and 3c. How does the length of the overlapped edges (or the width of GNRs) affect the edge states?**
- 5. The opposite edges (with red and blue dots at the center, respectively) in Fig. 3a appear similar, while in Fig. 3b, the opposite edges appear differently, as well as Fig. 3c. How to explain the difference at the opposite edges?**
- 6. By changing twist angles, the length of the overlapped edges will vary. How does the length and twist angle affect the edge states?**
- 7. As the authors claimed that the model suggests that the interlayer distance will change the interlayer electrostatic potential and edge spin distribution. This seems pretty obvious, but the experimental realization is a different story. I believe further discussions on this point are needed.**
- 8. A general discussion on how the twist angle of the junctions and the width of GNRs affect the edge states would make the manuscript more complete and broader interesting.**

Reviewer #2 (Remarks to the Author):

In their manuscript, D. Wang et al. present a detailed experimental and theoretical study of the electronic and magnetic structure of twisted bilayers of bottom-up zigzag graphene nanoribbons (ZGNRs). In their theoretical calculations, the authors show that

significant bandgap renormalization can take place in bilayer stacks of 6-ZGNRs due to the interlayer interaction of zigzag edge states. Unlike analogous twisted bilayer structures in 2D, the resultant band structure of twisted bilayer 6-ZGNRs also depends significantly on the registry between the layers (referred to as in-plane stacking offsets in the text). These stacking offsets can influence the symmetry of the interfacial electronic and magnetic structure, and can in theory lift the spin degeneracy of the zigzag states (i.e., break sublattice symmetry). On the experimental side, the authors conduct scanning tunneling spectroscopy on a range of twisted bilayer 6-ZGNR (6-atom-wide ZGNRs) structures, focusing mainly on a group of three 90 degree rotated variants. The authors propose three potential interfacial chemical structures whose theoretical density of states resemble the spectra collected on the associated experimental structure.

The notion of emergent electronic structure in twisted bilayers is a subject of intense research in condensed matter and materials science. Bridging this concept with the atomic-scale precision of bottom-up graphene nanoribbons has the potential to attract a broad audience from different sectors of materials research. It is conceivable that this study will motivate additional innovative research at the interface of these two subfields. With that being said, I believe the authors' manuscript (in its current form) leaves several open questions (particularly with regards to the interpretation of the experimental data), and invites potential alternative explanations to those presented by the authors. Below I outline several significant issues with the paper as it currently stands. The authors will need to address these points before I can recommend publication of their article in Nature Communications.

1. The most significant issue with the manuscript is the ambiguity about the overlapping interfacial structures in the experimental GNRs studied. Given the dependence of the electronic structure on the in-plane stacking offsets, having confidence in the experimental stacking offsets is crucial in order to compare the experiment to theory. Assigning the experimental structure should, in principle, be possible with bond-resolved imaging or referencing the undulations in the zigzag edge structure.
2. Related to point 1, the GNRs appear to be highly defective. While this isn't an issue per se, the presence of defects in the overlapping GNR region significantly affects interpretation of the data. For example, Figs. 2i-k shows the reversible dependence of the electronic structure on the interlayer stacking offsets. However, there appears to be several defects nearby in the underlying GNR, so the authors could simply be measuring the variable influence of a nearby defect and not the effect of stacking offsets.
3. Related to point 1 and 2, there are some structures with inexplicable electronic asymmetry that suggests the presence of a defect. For instance, experimental structures A and B should have 4- and 2-fold symmetry, respectively, based on the proposed model structures in Figs. 3d,e. However, there is clearly an asymmetry in the LDOS as seen in the insets of Figs. 3a,b. Can the authors reasonably rule out the possibility that this asymmetry is due to defects? If not defects, what is the cause of the asymmetry?
4. Why were 6-ZGNRs chosen as a model system for testing twisted GNR bilayers. It is known that the zigzag edge states are heavily obscured by interactions with the underlying Au(111) substrate. It would appear that armchair or chiral GNRs would be a preferable platform to test the angle-dependence of interlayer hybridization since they do not suffer from this issue with the Au(111) substrate.
5. Since the zigzag edge states exist within flat bands in their native state, can the authors truly conclude that the interlayer interactions and/or moiré potential are causing further flattening of the bands? The focus of the discussion seems to be more on bandgap renormalization than anything else.
6. Are the authors able to estimate the magnitude of the external electric field from their theoretical calculations and compare it to those presented in ref. 20? If they are comparable, then this would better justify this conclusion.
7. Line 140 says that the TBZGNR ". . . cannot be treated as a quantum dot . . ." However, the opposite seems to be true. Can the authors elaborate on this point?
8. The authors speculate that the smaller band gap of the stacked 6-ZGNRS compared to a single 6-ZGNR on NaCl is due to charge redistribution. Is it possible that increased Thomas-fermi screening in the stacked structure is to blame for the smaller gap?

9. In Fig. 3D, it is not obvious that the stacked and native bandgaps are different. A peak-to-peak measurement of the gap would suggest the gaps are comparable. How are the authors defining the gap in this case?

Reviewer #3 (Remarks to the Author):

The article “Twisted bilayer zigzag-graphene nanoribbons with stacking offset-tunable edge states” by Wang et al. is an interesting and important work that reports new features that emerge in twisted bilayers of 1D graphene nanoribbons. In particular, edge states depend not only on the twist angle (like they do in twisted bilayers of 2D graphene sheets) but also on the stacking offset. The authors demonstrate good agreement between the edge states calculated using density functional theory and measured experimentally using scanning tunneling spectroscopy. This will be an important work in the field of twistrionics based on 1D systems. However, certain aspects of the paper are confusing and require additional clarification. I recommend that the article be published after the minor comments, below, are addressed.

1. It would be useful if the authors would elaborate on the statement “Manipulation of such edge states with tailored properties is a long-lasting interesting topic with potential applications in nanodevices.” For example, what are the tailored properties that the edge states exhibit? For what specific applications are the edge states useful? Providing such details will help the readers better understand the context of the work and appreciate the reported results.
2. In Fig. 1a, the edges of the blue and black ribbons in the schematic look to be twisted at an angle of 90° , but the blue and black arrows are not twisted at an angle of 90° —why is this? Also, there appear to be faded grey ribbons that are at twisted at different angles, but they are very grainy at the printed resolution and may make the schematic more confusing.
3. In Fig. 1i, it looks like the yellow region that highlights the moiré site with AA stacking is in the top right part of the overlap region; but in the corresponding electrostatic potential map in Fig. 1l, the moiré site looks like it is in the top left part of the overlap region. It would be useful to also highlight the moiré site in yellow in Fig. 1j-l so the readers can more clearly see the correspondence between the moiré sites with the yellow regions in Fig. 1g-i.
4. Please define the terms “middle-layer electrostatic potential” and “interlayer electrostatic potential”. Are they referring to the same concept? If yes, it would be clearer to use a single terminology.
5. Why does the STS at the edge of the monolayer ZGNR mimic the line shape of Au(111)? Shouldn't the edge of a monolayer ZGNR and a metallic Au(111) surface exhibit different PDOS?
6. The statement “In this sense, the TBZGNR junction cannot be treated as a quantum dot that is more similar to a single atom or molecule.” is not clear. I suggest that the sentence be rephrased. Is the author trying to state that the TBZGNR is or is not similar to a single atom or molecule?
7. Please define Δ^0 and Δ^1 .
8. In Fig. 3a, the lower edge gave energy gap values of 0.90 eV and 1.15 eV, while the upper edge gave values of 1.07 eV and 1.34 eV. Why do the gaps vary so much (0.17 and 0.29 eV) between the lower and upper edges?
9. The authors state “Compared to the gap values for the same type of ZGNR decoupled by a NaCl layer, $\Delta^0 = 1.5$ eV and $\Delta^1 = 1.9$ eV, the band gap in our case has diminished considerably.” Two things are different between Ref. 32 and this work: here, there is bilayer graphene and the substrate is Au(111), whereas in Ref. 32, there is monolayer graphene and the substrate is NaCl. How can the author decouple the effect of the layer number (bilayer vs. monolayer) and the effect of substrate (Au(111) vs. NaCl) on the resulting gap energies?

10. For the peaks indicated in red in Fig. 3b,c and 3e,f, why are the experimental peaks sharper and asymmetric compared to the calculated peaks (which are broader and more symmetric)? Along the same lines, the authors state that the calculated peaks match the experimental data nicely in terms of shape and decay behaviour, but the data do not seem to match very well in terms of shape and decay behavior, but only match roughly qualitatively.
11. The authors state that “TBZGNR junctions were constructed with twist angle θ well controlled by STM tip lateral manipulation (with accuracy less than 5°).” But the authors only report fabrication of TBZGNRs with twist angle $> 30^\circ$, as shown in Fig. 2d and S3a. Can TBZGNRs with smaller twist angle be formed using this approach?
12. What are the very bright yellow, blob-like regions of contrast in the STM images in Fig. S2?
13. In Fig. S6, the edges of the top GNR (indicated by black dashed lines) are not parallel—why not?
14. Line 37 and 86: Change “von-Hove” to “van-Hove”.

Point-by-point responses to the Reviewer comments

Here, we provide a point-by-point response to the reviewers' comments. We quote original comments in black *italic* typeface. Our responses are in regular black typeface. **Yellow shadow** marks the places we added or changed something in the manuscript/supplement. Our changes to the text are in **red**.

Response to Reviewer #1

In this work, the authors employ DFT and STM to investigate junctions of stacking zigzag graphene nanoribbons (ZGNRs) with various twist angles. The junctions were constructed by manipulating ZGNRs by STM. The main results include observation of edge states at the edges of the junctions, which are compared with DFT calculations. The experimental realization of the junctions is novel, and the results are interesting. The comparison between the measured STS and the DFT calculations seems reasonable.

However, the title of the manuscript is a little confusing and misleading since the work is on the junctions or the overlapped parts of two nanoribbons. The title sounds like the studied systems are the edges of bilayer nanoribbons made from twisted bilayer graphene. In principle, the junctions are closer to a quasi-zero-dimensional system than a one-dimensional one. So, the title should be changed to be more accurate.

Some other detailed comments and questions are also listed below. If the authors can address the comments and questions, I suggest publishing this manuscript in Nature Communications after revision.

Response: We thank reviewer #1 for notifying the importance of our work and considering publishing this manuscript in Nature Communications. We change the title to “Twisted bilayer zigzag-graphene nanoribbon **junctions** with stacking offset-tunable edge states” according to the reviewer’s suggestion to emphasize the system’s quasi-zero-dimensional nature. The point-by-point responses to the comments are as follow.

1. From the STM images in the manuscript, the stacking order (AA or AB) of the junctions can not be determined. Atomically resolved STM images of the junctions are needed to unambiguously determine the stacking order.

Response: We thank the reviewer for pointing out this very important issue. We employed a machine equipped with both non-contact atomic force microscope (nc-AFM) and STM to resolve the atomic configuration of the junction. Although this effort is very time consuming, we got some initial results regarding the structure and the scanning tunneling spectroscopy (STS, dI/dV). Because we cannot reproduce a junction which is exactly the same as those in the main text in such a short time and under the Covid-19 pandemic, we got a junction with a twist angle of 76° (TBZGNR-76). We compared the experimental dI/dV , the calculated DOS and the atomic configuration based on nc-AFM image of the TBZGNR-76 to demonstrate the stacking order we

determined by comparing the calculated DOS with the experimental dI/dV in Fig. 3 of the main text is correct. The new results are presented below as well as in Figure S11 in the supplementary information.

The new junction we studied has a twist angle of 76° . The STM image is shown in Fig. S11a. In order to determine the atomic structure, nc-AFM measurements were done on both the bottom and top 6-ZGNR as shown in Fig. S11b and c. By extending the model structures from AFM measurements we resolve the atomic model of this 76° TBZGNR junction and show it in Fig. S11d. By comparing the experimental dI/dV spectra (Fig. S11f) with the DFT-calculated PDOS (Fig. S11g) along a similar path across the junction, one can get good agreement between the experiment and the calculation, just like the data shown in the main text Figure 3. For example, the peaks just above the Fermi energy are strongest only at the edge, and the signal at the left edge is slightly stronger than that at the right edge, as highlighted by the red dashed lines. The relative energy positions of the peaks below the Fermi energy (highlighted by black and blue arrows) to those just above the Fermi energy (highlighted by red dashed lines) also agree qualitatively between experiment and calculation. Noteworthy, the as-constructed 76° junction also lacks either inversion or mirror symmetry, so asymmetric edge states were found both by experiment and calculation. Thus, the nc-AFM measurements further support our argument in the main text and emphasize the importance of stacking offset in the determination of edge state.

Fig. S11. (a) The STM image of a TBZGNR junction with a twist angle of 76° . Size: $7 \text{ nm} \times 10 \text{ nm}$. (b, c) Zoom-in AFM images of the bottom ZGNR (light blue) and top ZGNR (black), respectively. Size: $0.7 \text{ nm} \times 1 \text{ nm}$. The models are superimposed on the images. (d) TBZGNR junction with the stacking configuration obtained by extending the AFM-measured structures in (b) and (c). (e, f) 19 dI/dV spectra (f) taken across the top ribbon edges along the arrow direction shown in (e). (g) DFT calculated PDOS on 21 atoms along the path (colorful arrow) shown in model (d). The red dashed lines in (f) and (g) highlight the edge states just above Fermi energy.

New discussions about the nc-AFM measurements are added in the main text on page 12

“Based on further nc-AFM measurements on an asymmetric TBZGNR structure with twist angle 76° , the atomic structure of the junction was determined in an unambiguous way. The measured dI/dV signal across the junction also matches with the calculated PDOS using the same atomic model (Figure S11).”

In supplementary information, page 7, we add the section “Nc-AFM measurements on a 76° TBZGNR junction together with DFT calculation” to support our arguments.

“To unambiguously make a link between the structure of the junction and the observed edge state, we employed both STM and nc-AFM and got some initial results regarding the structure and DOS. The new junction we studied has a twist angle of 76° as shown in Fig. S11a. In order to determine the atomic structure, nc-AFM measurements were done on both the bottom and top 6-ZGNR as shown in Fig. S11b and c. By extending the model structures from nc-AFM measurements we resolve the atomic model of this 76° TBZGNR junction shown in Fig. S11d. By comparing the experimental dI/dV spectra (Fig. S11f) with the DFT-calculated PDOS (Fig. S11g) along a similar path across the junction, one can get good agreement between the experiment and the calculation, just like the data shown in the main text Figure 3. For example, the peaks just above the Fermi energy are strongest only at the edge, and the signal at the left edge is slightly stronger than that at the right edge, as highlighted by the red dashed lines. The relative energy positions of the peaks below the Fermi energy (highlighted by black and blue arrows) to those just above the Fermi energy (highlighted by red dashed lines) also agree qualitatively between experiment and calculation. Noteworthy, the as-constructed 76° junction also lacks inversion or mirror symmetry, so asymmetric edge states were found both by experiment and calculation. Thus, the nc-AFM measurements further support our argument in the main text and emphasize the importance of stacking offset in the determination of edge state.”

2. For 0 twist angle (or parallel) bilayer GNRs, what are the possible edge configurations? How does the edge-to-edge interaction change the geometry of the edges?

Response: When the twist angle $\theta=0^\circ$ (parallel), two layers of ZGNRs most probably exhibit AA or AB stacking just like the non-twist bilayer graphene (Phys Rev B 78, 045404 (2008); Carbon 50, 784-790 (2012); Chin Phys Lett 28, 047304 (2011)). Other stackings are transitional configurations between the two. We didn't intend to list all possible edge configurations of parallel bilayer GNRs, but only listed two typical cases of AA- and AB-stacking to highlight the varying edge states with the zero-twist angle. However, edge-to-edge interactions bring changes to the electronic structure (e.g. bandgap), but are not strong enough to change the atomic geometry (e.g. stacking offset) of the edges. To clarify the atomic-edge configurations with AA- and AB-stackings, we added the following to the supporting information as Figure S12.

Figure S12. Top- (upper panels) and side-views (lower panels) of AA- (left) and AB-stacking (right) bilayer ZGNRS geometry. Grey: bottom ribbon, Blue: Top ribbon. In the top view of the AA-stacking, the top ribbon is shifted a bit for clear visualization. The p_z orbitals of edge carbon atoms were illustrated to help recognizing the atomic stacking on the edges.

In the case of AA stacking, one ribbon just sits directly on top of the other atom by atom. The edge-edge interaction is maximized because of the head-to-head p_z orbitals of the edge-carbon atoms, inducing an interlayer distance of 3.11 Å. In contrast, in the AB-stacking case, the edge atoms are not aligned, so the edge-edge interaction is not so strong as that in the AA case, inducing an interlayer distance of 3.28 Å. In the AB stacking case, each layer basically maintains its own intrinsic edge properties, as shown in Fig 1b.

3. In Fig. 2i and Fig. 2k, the twist angles appear different. Did the manipulation change the twist angle? If so, the manipulation process is not reversible.

Response: We thank the reviewer for pointing out this issue. Yes, after the manipulation, as shown in Fig. 2i-k, it seems that the twist angle changes a bit. Figures 2i-k were to demonstrate the ability to move the top ribbon back and forth relative to the bottom ribbon, as highlighted by the white arrows. In consequence, the STS (Fig. 2l) shows that edge states changed a lot after the first manipulation but were reproduced after manipulating it back. The reversibility is more to the edge states rather than the twist angle. These direct measurements on the same TBZGNR junction demonstrated the effect of stacking offset on edge states and its tunability.

For clarification, we changed the caption of Figure 2 as the following.

“...demonstrating the reversible manipulation ...” “The relative motion of the top ribbon is highlighted by the white arrows.”

We changed the discussion in the main text on Page 7, as follow.

“Once TBZGNR junctions were built, further manipulations on the top ribbon can still be achieved in both directions relative to the bottom ribbon...”

4. The different parts of the overlapped edges of the top GNRs show different edge states, e.g., middle of the edge vs. corners of the junction, as shown in Figs. 3b and 3c. How does the length of the overlapped edges (or the width of GNRs) affect the edge states?

Response: Thank the reviewer for raising this interesting question. Longer overlapped edges or wider GNRs produce bigger overlapped regions, where there are much more possibilities for edge configurations. As the bandgaps decrease fast when the width of GNRs becomes wider, there is less room to tune the edge states in wide GNRs (PRL 99, 186801 (2007)). In consequence, the edge-states analysis will be more complicated for wider overlapped edges. Our work used 6-ZGNR as a prototype to disclose the dominance of stacking angle and offset to these twisted one-dimensional systems.

We performed calculations on two more cases of narrower and wider GNRs compared to the 6-ZGNR with a twist angle of 90° . Narrower 4-GNRs show fewer changes on overlapped edges (along the edge direction), while wider 8-GNRs show more abundant changes (shown in the following figure).

We add following **Figure S13** to the supplementary information:

Figure S13: Configurations and PDOS on edge atoms of 4-ZGNR (a,b) and 8-ZGNR (c,d) with a twist angle of 90° . It is clear that in the 4-ZGNR there are some changes on edge states but not very pronounced, while in the 8-ZGNR the edge-states changes are much more abundant than those in 4-ZGNR and in 6-ZGNR, suggesting that the wider ZGNRs will produce more complicated the overlapped configurations and more abundant edge states.

We add following discussion to the main text on Page 5:

“Calculated PDOS on edge atoms in narrower 4-ZGNRs and wider 8-ZGNRs are shown in Figure S13, suggesting that the wider ZGNRs will produce more complicated overlapped configurations and more abundant edge states.”

5. *The opposite edges (with red and blue dots at the center, respectively) in Fig. 3a appear similar, while in Fig. 3b, the opposite edges appear differently, as well as Fig. 3c. How to explain the difference at the opposite edges?*

Response: We assume the reviewer is referring to that in Fig. 3b there is a tiny bright feature close to the bottom edge of the top ribbon. This feature most probably originates from a tiny stress difference between the two edges. The important point is that, as can be seen from the dI/dV mapping image shown in Fig. 2g, this feature does not give any additional contribution to the edge states. In Fig. 3c, the bright feature mostly lies at the top left corner of the junction, which could be due to a local bending difference between the top and the bottom edges. According to our strain analysis in the supplementary information, no considerable effect will be introduced.

6. *By changing twist angles, the length of the overlapped edges will vary. How does the length and twist angle affect the edge states?*

Response: If we fix the width of the ribbon, as one changes the twist angles, the length of the overlapped edges will vary as the reviewer pointed out. As demonstrated in Fig. 1c-1e, when we fix the moiré site at the center of the TBZGNR junction (the configuration with the highest symmetry), the edge state moves closer to zero energy as the twist angle increases from 30° to 60° (or the length of the overlapped edges decreases). We note that this conclusion is valid for the junctions with such high symmetry but is not guaranteed for the other cases. Extensive explorations for a comprehensive understanding of this issue would be a nice following-up paper inspired by this work.

We add the following discussion to the main text on Page 4:

“Figure 1c-e clearly show that the edge states (solid curve, shown as VHS in PDOS) of TBZGNR junctions shift towards zero energy with an increasing twist angle θ when the moiré site locates at the junction center.”

7. *As the authors claimed that the model suggests that the interlayer distance will change the interlayer electrostatic potential and edge spin distribution. This seems pretty obvious, but the experimental realization is a different story. I believe further discussions on this point are needed.*

Response: We thank the reviewer for the good suggestions, and we added further discussions in the main text on Page 14 to inspire further study.

“One possible experimental realization is fabricating TBZGNR junctions encapsulated between two insulating layers, i.e., boron nitrides. The out-of-plane stacking offset can be adjusted by tuning the insulating layers.”

8. *A general discussion on how the twist angle of the junctions and the width of GNRs affect the edge states would make the manuscript more complete and broader interesting.*

Response: We have added the above-mentioned new calculation results and texts to the supplementary information and the main text, which contribute to the reviewer-recommended general discussion in appropriate places. In addition, we add the following sentences to the main text on Page 12.

“Twist angles other than 90° and other widths of GNRs produce longer or shorter overlapped edges, where we believe the symmetry on interlayer electrostatic potential still affects the edge states. However, the quantitatively calculational and experimental measurements need further explorations.”

Response to Reviewer #2

In their manuscript, D. Wang et al. present a detailed experimental and theoretical study of the electronic and magnetic structure of twisted bilayers of bottom-up zigzag graphene nanoribbons (ZGNRs). In their theoretical calculations, the authors show that significant bandgap renormalization can take place in bilayer stacks of 6-ZGNRs due to the interlayer interaction of zigzag edge states. Unlike analogous twisted bilayer structures in 2D, the resultant band structure of twisted bilayer 6-ZGNRs also depends significantly on the registry between the layers (referred to as in-plane stacking offsets in the text). These stacking offsets can influence the symmetry of the interfacial electronic and magnetic structure and can in theory lift the spin degeneracy of the zigzag states (i.e., break sublattice symmetry). On the experimental side, the authors conduct scanning tunneling spectroscopy on a range of twisted bilayer 6-ZGNR (6-atom-wide ZGNRs) structures, focusing mainly on a group of three 90-degree rotated variants. The authors propose three potential interfacial chemical structures whose theoretical density of states resemble the spectra collected on the associated experimental structure.

The notion of emergent electronic structure in twisted bilayers is a subject of intense research in condensed matter and materials science. Bridging this concept with the atomic-scale precision of bottom-up graphene nanoribbons has the potential to attract a broad audience from different sectors of materials research. It is conceivable that this study will motivate additional innovative research at the interface of these two subfields.

With that being said, I believe the authors' manuscript (in its current form) leaves several open questions (particularly with regards to the interpretation of the experimental data), and invites potential alternative explanations to those presented by the authors. Below I outline several significant issues with the paper as it currently stands. The authors will need to address these points before I can recommend publication of their article in Nature Communications.

Response: We thank the reviewer for notifying the significance of our work and consideration of recommending the publication of our article in Nature Communications.

1. The most significant issue with the manuscript is the ambiguity about the overlapping interfacial structures in the experimental GNRs studied. Given the dependence of the electronic structure on the in-plane stacking offsets, having confidence in the experimental stacking offsets is crucial in order to compare the experiment to theory. Assigning the experimental structure should, in principle, be possible with bond-resolved imaging or referencing the undulations in the zigzag edge structure.

Response: We thank the reviewer for the good suggestion. As bond resolved imaging needs a CO on the tip and we perform a lot of tip manipulation in our experiment, nc-AFM was chosen to link the real atomic structure of the junction with its density of state on the edge. Because we cannot reproduce a junction which is exactly the same as those in the main text in such a short time and under the Covid-19 pandemic, we got a junction with a twist angle of 76° (TBZGNR-76). We compared the experimental dI/dV, the calculated DOS and the atomic configuration based on nc-AFM image of the TBZGNR-76 to demonstrate the stacking order we determined by comparing the calculated DOS with the experimental dI/dV in Fig. 3 of the main text is correct. **The new results are shown below as well as in Figure S11.**

In order to determine the atomic structure, nc-AFM measurements were done on both the bottom and top 6-ZGNR as shown in Fig. S11b and c. By extending the model structures from nc-AFM measurements we can obtain the atomic model of this 76° TBZGNR junction as shown in Fig. S11d. By comparing the dI/dV spectra in experiment (Fig. S11f) with DFT calculated PDOS (Fig. S11g) along similar path across the junction, one can get good agreement between the experiment and theory, just like the data shown in the main text Figure 3. For example, the peak just above Fermi energy are strongest at the edge and the signal at left edge is stronger than the right edge, as highlighted by the red dashed lines. The relative energy positions of the peaks below the Fermi energy (highlighted by black and blue arrows) to those just above the Fermi energy (highlighted by red dashed lines) also agree qualitatively between experiment and calculation. Noteworthy, the as-constructed 76° junction also lacks either inversion or mirror symmetry, so asymmetric edge states were found both by experiment and calculation. Thus, the nc-AFM measurements further support our argument in the main text and emphasize the importance of stacking offset in the determination of edge state. However, as this type of characterize is extremely hard and time consuming, we only took one example to testify our proposal.

Fig. S11. (a) The STM image of a TBZGNR junction with a twist angle of 76° . Size: $7 \text{ nm} \times 10 \text{ nm}$. (b, c) Zoom-in AFM images of the bottom ZGNR (light blue) and top ZGNR (black), respectively. Size: $0.7 \text{ nm} \times 1 \text{ nm}$. The models are superimposed on the images. (d) TBZGNR junction with the stacking configuration obtained by extending the AFM-measured structures in (b) and (c). (e, f) 19 dI/dV spectra (f) taken across the top ribbon edges along the arrow direction shown in (e). (g) DFT calculated PDOS on 21 atoms along the path (colorful arrow) shown in model (d). The red dashed lines in (f) and (g) highlight the edge states just above Fermi energy.

New discussions about the nc-AFM measurements are added in the main text on page 12

“Based on further nc-AFM measurements on an asymmetric TBZGNR structure with twist angle 76° , the atomic structure of the junction was determined in an unambiguous way. The measured

dI/dV signal across the junction also matches with the calculated PDOS using the same atomic model (Figure S11).”

In supplementary information, page 7, we add the section “Nc-AFM measurements on a 76° TBZGNR junction together with DFT calculation” to support our arguments.

“To unambiguously make a link between the structure of the junction and the observed edge state, we employed both STM and nc-AFM and got some initial results regarding the structure and DOS. The new junction we studied has a twist angle of 76° as shown in Fig. S11a. In order to determine the atomic structure, nc-AFM measurements were done on both the bottom and top 6-ZGNR as shown in Fig. S11b and c. By extending the model structures from nc-AFM measurements we resolve the atomic model of this 76° TBZGNR junction shown in Fig. S11d. By comparing the experimental dI/dV spectra (Fig. S11f) with the DFT-calculated PDOS (Fig. S11g) along a similar path across the junction, one can get good agreement between the experiment and the calculation, just like the data shown in the main text Figure 3. For example, the peaks just above the Fermi energy are strongest only at the edge, and the signal at the left edge is slightly stronger than that at the right edge, as highlighted by the red dashed lines. The relative energy positions of the peaks below the Fermi energy (highlighted by black and blue arrows) to those just above the Fermi energy (highlighted by red dashed lines) also agree qualitatively between experiment and calculation. Noteworthy, the as-constructed 76° junction also lacks inversion or mirror symmetry, so asymmetric edges states were found both by experiment and calculation. Thus, the nc-AFM measurements further support our argument in the main text and emphasize the importance of stacking offset in the determination of edge state.”

2. Related to point 1, the GNRs appear to be highly defective. While this isn't an issue per se, the presence of defects in the overlapping GNR region significantly affects interpretation of the data. For example, Figs. 2i-k shows the reversible dependence of the electronic structure on the interlayer stacking offsets. However, there appears to be several defects nearby in the underlying GNR, so the authors could simply be measuring the variable influence of a nearby defect and not the effect of stacking offsets.

Response: We believe the reviewer is referring to the bright-dot features on the overlapped edges. We checked the on-edge bright-dot features carefully and exclude them to be defects or adatom. We attribute them to different stress states of edges. This can be evidenced by Fig 2f and g. There is an on-edge bright-dot feature in Fig. 2f but the dI/dV mapping in Fig. 2g shows no additional signal.

We further checked the dI/dV on and off the bright-dot feature for the junction in Fig. 2j, as shown in Figure R1 below. The dI/dV on the bright feature (blue dot) shows no significant difference from the other sites on the same edge (red and black dots) as well as the opposite edge (green dot). Moreover, this bright feature disappears after manipulating back, as shown in Fig. 2k. It demonstrates that the bright feature is not a defect or an adatom. We add the corresponding discussion as a new section in the Supporting Information on page 6, “Exclusion of the influence of bright protrusions on the edge state”.

“As there are some bright protrusions on Fig. 2f and 2j, it is a necessary to check they has no relation with the edge states we discuss in the paper. By checking the dI/dV mapping in Fig. 2g we found no additional signal belongs to the protrusion shown in Fig. 2f at the edge state energy -40 mV. From the tip manipulation of the top ribbon shown in Fig. 2i-2k, we learned the bright feature on the right edge of top ribbon appears for the first manipulation (Fig. 2j) and disappears again (Fig. 2k) after manipulating back. As our tip is far from the junction during the manipulation, it is very unlikely to be adatom or carbon atom defect. We propose the protrusions we observed to be a tiny stress states at the edges which, as we demonstrated already in previous section, will not lead to great impact (the top ribbon edge at the junction is still very straight and has a strain less than 1%).”

Figure R1 Left: same with Fig. 2j but labeled three positions where additional spectra were taken. Right: STS taken on point 2 as well as points A, B and C as indicated in the left image.

3. Related to point 1 and 2, there are some structures with inexplicable electronic asymmetry that suggests the presence of a defect. For instance, experimental structures A and B should have 4- and 2-fold symmetry, respectively, based on the proposed model structures in Figs. 3d,e. However, there is clearly an asymmetry in the LDOS as seen in the insets of Figs. 3a,b. Can the authors reasonably rule out the possibility that this asymmetry is due to defects? If not defects, what is the cause of the asymmetry?

Response: We thank for the reviewer raising the question regarding the asymmetry of the peak. First of all, a defect-bound state is not guaranteed to be must asymmetry, i.e. the Kondo bound state is usually symmetric at this energy scale. Thus, the defect is not the only possible origin of the asymmetry. Second, this feature is very unlikely due to defect or adatom, as stated in the previous response to comment #2, which is summarized to be: 1) The dI/dV mapping signal at the corresponding energy -40 mV in Fig. 2g and the dI/dV spectra in Figure R1 show no clear decay

effect when we measured from the bright feature to the surroundings. 2) From Fig 2j to Fig 2k, the bright feature disappears again after manipulating the ribbon back, excluding the possibility of a defect or an adatom.

Moreover, although our models include 4- and 2- fold symmetry, as soon as one considers the Au substrate or the real extended junction structure, these symmetries will be broken. In reality, when we took the STS data at the edge, there is very little but non-zero probability that the tip is tunneling to the bulk DOS (i.e. into the C atoms next to the edge). These contributions will make the peak more asymmetric as the bulk bands are below the edge bands.

4. Why were 6-ZGNRs chosen as a model system for testing twisted GNR bilayers. It is known that the zigzag edge states are heavily obscured by interactions with the underlying Au(111) substrate. It would appear that armchair or chiral GNRs would be a preferable platform to test the angle-dependence of interlayer hybridization since they do not suffer from this issue with the Au(111) substrate.

Response: We thank the reviewer for the good suggestion. The AGNR is a very nice platform one could try in future experiments due to its less interaction with the underlying Au. However, the most important reason we chose ZGNR is the novel magnetic ordering at the ribbon edges. With the twisted bilayer ZGNR junctions, we aimed not only to decouple the top-layer ZGNR from the Au substrate but also to tune the magnetic edge states by changing the twisting angle and stacking offset. We obtained more insights into the mechanism of manipulating the edge states, which is hopefully applicable in future spintronics. Recently, a theoretical study just predicts the TBZGNR junction is a nice platform for spin-polarizing electron beam splitter (Phys. Rev. Lett. 129, 037701, 2022).

5. Since the zigzag edge states exist within flat bands in their native state, can the authors truly conclude that the interlayer interactions and/or moiré potential are causing further flattening of the bands? The focus of the discussion seems to be more on bandgap renormalization than anything else.

Response: It is true that the changing of the energy gap in the three model structures can also be viewed as bandgap renormalization. The newly appeared peaks near zero can be interpreted as the DOS at the new top valence band in Fig. 4b model B and model C. However, here we mainly want to compare the band dispersion and spin polarization difference in the stacked structure with the pristine GNR. In pristine GNR, the edge band is flat in the momentum space $2\pi/3 \leq |k| \leq \pi$ (the wavenumber k is normalized by the primitive translation vector of the ZGNR). This can also be seen in the folded band structure (Figure 4b pristine GNR, edge bands near -0.1 eV). When comparing this data with the edge bands in the stacked structure model B and model C, we see the bands shift towards zero and become flatter (less dispersive) in model B. The band near zero even splitting and even flatter in model C than in model B. By comparing the detailed band shifting and dispersion along the K space, we can understand more about what could happen to the edge band when we stack one ribbon on top of the other.

We revised the following sentence in the main text on page 9 to:

“We attribute this band gap reduction to the energy bands renormalization mainly caused by the

interlayer electron hopping-induced charge redistribution between the two ZGNR layers, which did not occur when the ribbon was decoupled by a NaCl layer.”

6. *Are the authors able to estimate the magnitude of the external electric field from their theoretical calculations and compare it to those presented in ref. 20? If they are comparable, then this would better justify this conclusion.*

Response: Thank the reviewer for raising this helpful justification. Although explicit calculations of the inter-layer potential are not straightforward because of the atomically distributed electrostatic potential, we can approximately read the potential difference within the overlapped region, ~ 0.5 eV, from Figure 1j-l. Considering the width of 6-ZGNR is about 10 \AA , the estimated differential electric field is $\sim 0.05 \text{ V/\AA}$, which is the same magnitude as shown in Figure 3c of ref. 20. Moreover, we notice that the number of 0.5 eV is also comparable to the estimated differential electrostatic potential in the overlapped wider, armchair GNR junctions in ref. 34.

We add the following sentence to the main text on Page 12.

“By extracting the potential difference within the overlapped region from Figure 11, the estimated differential electric field is $\sim 0.05 \text{ V/\AA}$, which is comparable with that predicted in reference 20.”

7. *Line 140 says that the TBZGNR “. . . cannot be treated as a quantum dot . . .” However, the opposite seems to be true. Can the authors elaborate on this point?*

Response: We understand the reviewer’s concern about the cross of the ZGNRs to be dots. Actually, the idea here is closer to the flat bands in magic-angle bilayer graphene: by overlapping two periodic materials, the band structures evolve as the twist angle changes and finally form flat bands – spatially localized electronic states. The key feature of flat bands is that they result from the evolution of intrinsic band structures and is highly tunable by parameters like twist angle. In addition, in cases of the twist angle becoming smaller or wider ZNGRs being used to form junctions, the overlapped edges will be extensive to be less like “dots”.

To avoid distracting the readers from our key idea, we decide to delete this sentence to avoid any possible misunderstanding.

8. *The authors speculate that the smaller band gap of the stacked 6-ZGNRS compared to a single 6-ZGNR on NaCl is due to charge redistribution. Is it possible that increased Thomas-fermi screening in the stacked structure is to blame for the smaller gap?*

Response: We thank the reviewer for raising this important consideration. From ref. PRL 99, 186801 (2007) we learned the dynamically screened Coulomb interaction (W) and dressed Green’s function (G) correction can greatly modify the energy gap calculated only by L(S)DA and describe the system more precise. Thus, if the quasiparticle (QP) on top layer is further screened by the bottom layer GNR or even by the Au(111) surface, the QP energy gap will be modified accordingly. The electron charge e placed at the distance d from a bulk conductor leads to a dipole potential evolving as $2ed^2/r^3$ at very large in-plane distances $r \gg d$, which is much weaker than the original, unscreened Coulomb potential, e/r . Thus, if the electrons at the edge of the top nanoribbon satisfy this criteria, electron-electron interaction will be decreased due to the Thomas-Fermi screening.

According to a recent report (Nature Communications 11, 2339, 2020), the Thomas-Fermi

screening starts to influence the e-e interaction only the distance between graphene and conductor smaller than a critical value $0.03\epsilon D$, where ϵ is the dielectric's permittivity, $D \approx 1/\sqrt{n}$, and n is the carrier density. The researchers observed an increased e-e mean free length when the BN thickness d become thinner (1.3 nm) due to the screening effect by the conductor.

In our experiment, we have very similar geometry, Top GNR/Bot GNR/Au(111), with comparable parameters. 1) the top ribbon is around $d=0.4$ nm above the Au(111) surface with a bottom ribbon in between. 2) The carrier density n for the bottom conduction band of the top ribbon can be chosen as 10^{12} cm^{-2} (Eur. Phys. J. B. 87, 50, 2014). 3) the dielectric's permittivity ϵ of the bottom GNR can be chosen as $6\epsilon_0$ (Phys. Rev. B 94, 045318 (2016), ϵ_0 is the vacuum permittivity and we define as 1 here). Thus, the distance between graphene nanoribbon and gold substrate $d=0.4$ nm which is smaller than the critical value $0.03\epsilon D=1.8$ nm. Even we consider our case as a Top GNR/Vacuum/Bot GNR heterostructure (ignore the effect of Au totally and view bot GNR as a metal), one gets a value $d=0.2$ nm which is also smaller than $0.03\epsilon_0 D=0.3$ nm. Thus, similar screen effects are expected as in the previous reports. In their case, the e-e mean free path increased from hundred nanometers to almost 1 micrometer due to the enhanced Thomas-Fermi screening. In other words, the screening effect play more important role for electrons with a separation of hundreds of nanometers. However, the graphene nanoribbon has a width of only 1 nm at which length scale the edges electrons still interacting a lot. If we assume the edge electrons separation $r=1 \text{ nm} \gg d=0.5 \text{ nm}$ **which is obviously not the case**, we can use the screened potential $2ed^2/r^3$ at $r_0=1 \text{ nm}$ in comparison with the unscreened potential e/r at $r=r_0$. A simple calculation shows the electron potential only reduced to half for the screened case at this short distance. But in reality this value should be bigger because we use the wrong potential equation at small distance. In this sense, we learned the Thomas-Fermi screening can dominate at larger length scale ($> 100 \text{ nm}$) but not guaranteed bigger enough to reduce the ribbon energy gap as we observed. More accurate model needs to be developed to give a quantified result. **In any case, we list this possible origin of bandgap reduction in the main text on page 9 to stimulate more study on this direction.**

“In addition, we can't easily exclude further bandgap renormalization mechanism such as Thomas-Fermi screening when including the effect of the Au (111) surface³⁹.”

9. In Fig. 3D, it is not obvious that the stacked and native bandgaps are different. A peak-to-peak measurement of the gap would suggest the gaps are comparable. How are the authors defining the gap in this case?

Response: We thank the reviewer for pointing out this unclarity. In the following and the main text, we update Fig. 3d by aligning the valence-band peak of the three PDOS curves (left black dashed line). It is more visually obvious that the conduction-band peaks of model-A edges are lower in energy than that of intrinsic ZGNR as calculated, indicating narrower bandgaps in model A. That being said, we defined the gap by measuring the energy difference between the peak positions in the PDOS.

Updated Fig. 3d.

Response to Reviewer #3

The article “Twisted bilayer zigzag-graphene nanoribbons with stacking offset-tunable edge states” by Wang et al. is an interesting and important work that reports new features that emerge in twisted bilayers of 1D graphene nanoribbons. In particular, edge states depend not only on the twist angle (like they do in twisted bilayers of 2D graphene sheets) but also on the stacking offset. The authors demonstrate good agreement between the edge states calculated using density functional theory and measured experimentally using scanning tunneling spectroscopy. This will be an important work in the field of twistrionics based on 1D systems.

However, certain aspects of the paper are confusing and require additional clarification. I recommend that the article be published after the minor comments, below, are addressed

We thank the reviewer for acknowledging the significance of our work and for recommending publishing our manuscript after we address the comments. The responses to the comments are as follow.

1. It would be useful if the authors would elaborate on the statement “Manipulation of such edge states with tailored properties is a long-lasting interesting topic with potential applications in nanodevices.” For example, what are the tailored properties that the edge states exhibit? For what specific applications are the edge states useful? Providing such details will help the readers better understand the context of the work and appreciate the reported results.

Response: We appreciate the reviewer’s suggestion and elaborate on the corresponding statement more specifically. **We updated the sentence in the main text on Page 2** as: **“Manipulation of such edge states with tailored properties, such as antiferromagnetic semiconductor to ferromagnetic half-metal transition²⁰, spin-splitting of dopant edge states²¹ and topological order²², is a long-lasting interesting topic with potential applications in nanodevices, i.e. spintronics^{23,24} and quantum bits²⁵.”**

2. In Fig. 1a, the edges of the blue and black ribbons in the schematic look to be twisted at an angle of 90°, but the blue and black arrows are not twisted at an angle of 90°—why is this? Also, there appear to be faded grey ribbons that are at twisted at different angles, but they are very grainy at the printed resolution and may make the schematic more confusing.

Response: Figure 1a, as the reviewer noticed, is a schematic, however, not only for the 90° angle but also for other angles have been manipulated on the top black ribbon. This explains why the blue and black arrow does not indicate 90° (it is intended to represent a general twist angle) and the faded grey ribbons (they are the trace when the top ribbon rotates). To better illustrate the intention of the schematic, we added two arrows indicating the top-ribbon rotation and a grey dashed line perpendicular to the blue arrow indicating the 90° and non-90° situations. We note that the generality of twist angles in Fig. 1a is important so as to lead the following Figs. 1b-e.

The updated Fig. 1a is as the following for the reviewer’s convenience.

Updated Fig. 1a

3. In Fig. 1i, it looks like the yellow region that highlights the moiré site with AA stacking is in the top right part of the overlap region; but in the corresponding electrostatic potential map in Fig. 1l, the moiré site looks like it is in the top left part of the overlap region. It would be useful to also highlight the moiré site in yellow in Fig. 1j-l so the readers can more clearly see the correspondence between the moiré sites with the yellow regions in Fig. 1g-i.

Response: We thank the reviewer for pointing out this unclarity. We corrected the orientation of Fig. 1l. The updated Fig. 1l is as the following for the reviewer’s convenience.

However, when we tried to highlight the moiré sites in yellow in Fig. 1j-l, it is very hard to visualize the shaded yellow highlights. We tried other color shadows but they influence the contrast of the color-bar gradience. Instead of adding color shadows, we added open stars on each moiré site in Figs. 1g-l, which helps a lot for the readers to notice the correspondence.

Updated Fig. 1l

4. Please define the terms “middle-layer electrostatic potential” and “interlayer electrostatic potential”. Are they referring to the same concept? If yes, it would be clearer to use a single terminology.

Response: Yes, they mean the same. We unify the terminology in the main text to interlayer electrostatic potential and give the corresponding definition in the caption of Fig. 1(j-l).

“DFT-calculated interlayer electrostatic potential (in the middle plane between the top and bottom GNRs) of three 90°-TBZGNRs...”

5. Why does the STS at the edge of the monolayer ZGNR mimic the line shape of Au (111)? Shouldn't the edge of a monolayer ZGNR and a metallic Au (111) surface exhibit different PDOS?

Response: To our best knowledge, this is due to the Au (111) surface hosting a very strong surface state. The ZGNR edge states are not observed mainly due to the strong expansion of the surface state of Au (111) in the out of plane direction. From the Figure 2 in Ref. *Phys. Rev. B* 85, 245440 we can clearly see the Au (111) surface state survive even 2 Å away from the surface. The apparent height of our monolayer ZGNR is just 1.85 Å. Thus, most of the ZGNR edge states are in the shadow of the Au (111) surface state, this is also the reason why the DOS at the monolayer edge mimics the surface state of Au (111), as already reported in the previous study (*Nature* 531, 489–492 (2016), Figure S4 C).

6. The statement “In this sense, the TBZGNR junction cannot be treated as a quantum dot that is more similar to a single atom or molecule.” is not clear. I suggest that the sentence be rephrased. Is the author trying to state that the TBZGNR is or is not similar to a single atom or molecule?

Response: We presume some readers may concern about the crossed ZGNR being zero-dimension systems. We would like to emphasize that the idea here is closer to the flat bands in magic-angle bilayer graphene: by overlapping two periodic materials, the band structures evolve as the twist angle changes and finally form flat bands – spatially localized electronic states. The key feature of flat bands is that they are resulted from the evolution of intrinsic band structures and is highly tunable by parameters like twist angle. In addition, in cases of the twist angle becoming smaller or wider ZNGRs being used to form junctions, the overlapped edges will be extensive to be less like “dots”.

To avoid distracting the readers from our key idea, we decide to delete this sentence to avoid any possible misunderstanding.

7. Please define Δ^0 and Δ^1 .

Response: We define Δ^0 and Δ^1 in the main text on Page 9 as:

“ Δ^0 and Δ^1 denote the direct band gap and the energy gap at the Brillouin zone boundary¹³.”

8. In Fig. 3a, the lower edge gave energy gap values of 0.90 eV and 1.15 eV, while the upper edge gave values of 1.07 eV and 1.34 eV. Why do the gaps vary so much (0.17 and 0.19 eV) between the lower and upper edges?

Response: We thank the reviewer for raising the question and this question is in fact non-trivial. Compared to the upper edge hosting gap values $\Delta^0 = 1.07$ eV and $\Delta^1 = 1.34$ eV, the lower edge gap values change to 0.17 eV (15.8%) and 0.19 eV (14%) correspondingly. The tiny gap difference most probably due to possible symmetry broken when we include the substrate in experiment.

9. The authors state “Compared to the gap values for the same type of ZGNR decoupled by a NaCl

layer, $\Delta^0 = 1.5 \text{ eV}$ and $\Delta^1 = 1.9 \text{ eV}$, the band gap in our case has diminished considerably.” Two things are different between Ref. 32 and this work: here, there is bilayer graphene and the substrate is Au(111), whereas in Ref. 32, there is monolayer graphene and the substrate is NaCl. How can the author decouple the effect of the layer number (bilayer vs. monolayer) and the effect of substrate (Au(111) vs. NaCl) on the resulting gap energies?

Response: We would like to clarify that in our work the bottom-layer ZGNR plays the same role as NaCl does in Ref. 37 (Nature 531, 489-492 (2016), labeled as Ref. 32 in the old version main text) - a buffer layer decoupling the top-layer ZGNR from the Au substrate. Since bottom-layer ZGNR is more conductive than NaCl, our measured bandgaps of top-layer ZGNR turn out to be comparable with but different from the reported in Ref 37. However, the twist angle and stacking offsets between two layers of ZGNR give out new physics for tuning edge states, which serves our key idea for this work.

10. For the peaks indicated in red in Fig. 3b,c and 3e,f, why are the experimental peaks sharper and asymmetric compared to the calculated peaks (which are broader and more symmetric)? Along the same lines, the authors state that the calculated peaks match the experimental data nicely in terms of shape and decay behaviour, but the data do not seem to match very well in terms of shape and decay behavior, but only match roughly qualitatively.

Response: We agree with the reviewer and replace the original sentence with “...match the experimental data qualitatively...” in the new version main text on Page 10. The calculations were performed with optimized freestanding models, so they gave electronic states contributed by individual edge atoms. In experiments, however, the STS data at the edge were obtained by collecting the tunneling signals from tip to the atom below it as well as its surroundings which depends on the tip sharpness. There is very little but non-zero probability that the tip is tunneling to the bulk DOS (i.e. into the C atoms next to the edge). This contribution will make the peak more asymmetric as the bulk bands are below the edge bands. Moreover, in experiment, the junction is not guaranteed symmetric when including the substrate and the extending region of the junction. The peak width in calculational PDOS here is simply numerical broadening. It is the peak position that is important. However, the bandgap is underestimated at the level of calculations in this work, so we decide to change the wording as the reviewer suggested.

11. The authors state that “TBZGNR junctions were constructed with twist angle θ well controlled by STM tip lateral manipulation (with accuracy less than 5°).” But the authors only report fabrication of TBZGNRs with twist angle $> 30^\circ$, as shown in Fig. 2d and S3a. Can TBZGNRs with smaller twist angle be formed using this approach?

Response: Unfortunately, with this approach TBZGNRs with twist angle smaller than 30° is very hard to fabricated. To build such a small twist angle junction, one will push the ribbon as a whole instead of just pushing one end of the top ribbon and rotating it on top of the bottom. As soon as the top ribbon reaches the top of the bottom ribbon, because of the very large overlapping area at small angle, the interaction between these two layers become larger. Then, the bottom ribbon will also move during the manipulation. This is the bottle neck which limits fabricating the small twist angle junctions.

We modified the sentence in the main text on Page 3 to make the discussion more precise.

“TBZGNR junctions were constructed with twist angle θ well controlled by STM tip lateral manipulation (with accuracy less than 5° and θ between 30° and 90°).”

12. What are the very bright yellow, blob-like regions of contrast in the STM images in Fig. S2?

Response: We are sorry about the contrast; it is over adjusted. The region with very high contrast on the right side of the ribbon is a big cluster made either by lots of precursors not involved in the Ullmann coupling or by some other solvents. We change the contrast in the new version of Fig. S2 to make them better visualized.

Updated Fig. S2

13. In Fig. S6, the edges of the top GNR (indicated by black dashed lines) are not parallel—why not?

Response: We thank the reviewer for the very carefully checking. That black line is a mistake during the production of the image. The edges are in fact parallel. We correct this in the new Figure S6.

(a)

Updated Fig. S6a

14. Line 37 and 86: Change “von-Hove” to “van-Hove”.

Response: We correct this error in the new version.

REVIEWERS' COMMENTS

Reviewer #1 (Remarks to the Author):

The authors have satisfactorily addressed my comments and revised the manuscript accordingly. Hence I recommend publication of the new version in Nature Communications.

Reviewer #2 (Remarks to the Author):

In their revised manuscript, D. Wang et al. have made significant changes to their manuscript and have addressed the issues I raised during my initial review. As stated in my first response, I believe that this work has significant potential to attract researchers in many subfields of solid-state physics and materials science, making it suitable for publication in Nature Communications. Therefore, I suggest the manuscript be published after the authors consider the minor correction suggested below (though my recommendation is not contingent upon them making any further corrections).

1. The authors have made compelling arguments that the unexpected bright spots in the overlapping regions are not due to defects or adatoms (as seen for example in Figs. 2f,j, and the insets for Fig. 3a,b). The authors propose strain states and/or asymmetry in the underlying Au(111) substrate as playing a significant role in this respect. To completely rule out the potential role of defects on the observed electronic structure, it would be useful to include STS on one of the ubiquitous "mouse-bite" defects on a single-layer 6ZGNR to show that it does not resemble the STS in the bilayer structures.

Reviewer #3 (Remarks to the Author):

The authors have sufficiently addressed all of my previous comments. I believe that the manuscript is now ready to be published in Nature Communications.

Point-by-point responses to the Reviewer comments

Here, we provide a point-by-point response to the reviewers' comments. We quote original comments in black *italic* typeface. Our responses are in regular black typeface. Yellow shadow marks the places we added or changed something in the manuscript/supplement. Our changes to the text are in red.

Response to Reviewer #1

The authors have satisfactorily addressed my comments and revised the manuscript accordingly. Hence I recommend publication of the new version in Nature Communications.

Response: We thank reviewer #1 for recommending publication of our manuscript in Nature Communications.

Response to Reviewer #2

In their revised manuscript, D. Wang et al. have made significant changes to their manuscript and have addressed the issues I raised during my initial review. As stated in my first response, I believe that this work has significant potential to attract researchers in many subfields of solid-state physics and materials science, making it suitable for publication in Nature Communications. Therefore, I suggest the manuscript be published after the authors consider the minor correction suggested below (though my recommendation is not contingent upon them making any further corrections).

Response: We thank reviewer #2 for suggesting our work publishing in Nature Communications after minor correction.

1. The authors have made compelling arguments that the unexpected bright spots in the overlapping regions are not due to defects or adatoms (as seen for example in Figs. 2f,j, and the insets for Fig. 3a,b). The authors propose strain states and/or asymmetry in the underlying Au(111) substrate as playing a significant role in this respect. To completely rule out the potential role of defects on the observed electronic structure, it would be useful to include STS on one of the ubiquitous "mouse-bite" defects on a single-layer 6ZGNR to show that it does not resemble the STS in the bilayer structures.

Response: We thank reviewer #2 again for the suggestion to include the STS of "mouse-bite" type defect to completely rule out the role of defect. During the preparation of the TBZGNR junctions, we maximally avoided those defect regions. The unexpected bright spots shown in Figs.

2f,j and Fig. 3b in the overlapping regions are very unlikely due to this type of defects because the bright spots appear in the middle of the top ribbon edge within the junction, which means they sit on the middle of the bottom ribbon. However, the “mouse-bite” type defect appears mostly on the edge of the bottom ribbon. For the bright spots shown in Fig. 3a, it is also very unlikely to be this type of defects because it requires the two top ribbon edges sit on two “mouse-bite” type defects exactly. We didn’t build such TBZGNR junction on purpose.

As most of our work focus on the prefect region of the ribbons, we did not take STS on such defects. Luckily, a seminal paper from some of our coauthors indeed included those data (Nature 531, 489, 2016). As seen in Figure S4 in the supplementary information of this reference (also shown below), the STS on the “mouse-bite” type defect (indicated by red triangle) and normal zigzag edge on Au(111) (indicated by black triangle) show little difference. Most importantly, the lineshape of the STS on “mouse-bite” type defect does not resemble the STS in the bilayer structure, which we have shown in Figure 3(a-c) where either clear gap feature or in-gap states were observed. Thus, we can safely exclude that the bright features in Figs. 2f,j and the insets for Fig. 3a,b are this type of defect.

Fig. S4. dI/dV spectra recorded at different positions within a partially decoupled ZGNR. (A) Constant-current STM image ($T = 5$ K, $U = -0.25$ V, $I = 100$ pA) of a partially decoupled 6-ZGNR. Scale bar: 1nm. (B) Differential conductance (dI/dV) spectra recorded at decoupled zigzag edges (positions marked by colored dots in (A)). (C) Differential conductance (dI/dV) spectra recorded at zigzag edges on Au(111), a missing phenyl ring defect on Au(111), a missing phenyl ring defect on NaCl, and a protrusion defect of unknown nature (positions indicated by colored triangles in (A)). All spectra recorded at electronically decoupled zigzag edges (6-ZGNR on NaCl) show identical spectroscopic features deriving from the edge states, while the features are totally/partially missing for the other cases.

We added new discussion in the Supplementary Note 8 “Exclusion of the influence of bright protrusions on the edge state” to exclude the role of “mouse-bite” type defect.

“These bright protrusions are also excluded to be “mouse-bite” type defect on the edge of the single layer ZGNR because of two facts. Firstly, the bright protrusions shown in Figs. 2f,j and Fig. 3b appear in the middle of the top ribbon edge within the junction, which means they sit on the middle of the bottom ribbon. However, the “mouse-bite” type defect appears mostly on the edge of the bottom ribbon. Secondly, as seen in Figure S4 in the supplementary information of reference 4, the STS on the “mouse-bite” type defect (indicated by red triangle) does not resemble what we have shown in Figure 3(a-c) where either clear gap feature or in-gap states were observed. Thus, we can safely exclude that the bright features in Figs. 2f,j and the insets for Fig. 3a,b are this type of defect”

We added new discussion in the maintext:

“Other mechanism which can cause the DOS anomaly near zero energy such as defect state can also be ruled out (Supplementary Note 8).”

Response to Reviewer #3

The authors have sufficiently addressed all of my previous comments. I believe that the manuscript is now ready to be published in Nature Communications.

Response: We thank reviewer #3 for recommending publication of our manuscript in Nature Communications.